

# Chemical Characteristics of Marine Fine Aerosols over Sea and at Offshore Islands during Three Cruise Sampling Campaigns in the Taiwan Strait- Sea Salts and Anthropogenic Particles

Tsung-Chang, Li[1], Chung-Shin Yuan[1*], Chung-Hsuang Hung[2], Hsun-Yu Lin[3], Hu-Ching Huang[4],
Chon-Lin Lee[4]

[1] Institute of Environmental Engineering, National Sun Yat-sen University, Kaohsiung, 80424, Taiwan, R.O.C.
[2] Department of Safety, Health, and Environmental Engineering, National Kaohsiung First University of Science and Technology, Kaohsiung, 82445, Taiwan, R.O.C.
[3] China Steel Corporation, New materials R&D Department, Kaohsiung, 81233, Taiwan, R.O.C.
10 [4] Department of Marine Environment and Engineering, National Sun Yat-sen University, Kaohsiung, 80424, Taiwan, R.O.C.

*Correspondence to*: Chung-Shin Yuan (ycsngi@mail.nsysu.edu.tw)

**Abstract.** Marine fine aerosols were simultaneously collected over sea and at offshore islands during three cruise sampling campaigns to investigate the spatial distribution of atmospheric fine particles ($PM_{2.5}$) and the influences of sea salts and anthropogenic particles on the chemical characteristics of $PM_{2.5}$ in the Taiwan Strait. Field sampling results indicated that

$PM_{2.5}$ concentrations over sea were generally higher than those at the offshore islands, while the $PM_{2.5}$ concentrations in the daytime were commonly higher than those at nighttime. Moreover, the concentrations of $PM_{2.5}$ were generally higher than those of coarse particles ($PM_{2.5-10}$) with an exception of the winter cruise in 2014. Moreover, sea salts accounted for 6.5-11.1% and 11.0-13.5% of $PM_{2.5}$ at the offshore islands and over sea, respectively. The contributions of non-sea salt-water soluble ions (nss-WSI) to $PM_{2.5}$ at the offshore islands were obviously higher than those over sea, while the contributions of

ss-WSI for $PM_{2.5}$ at the offshore islands were much lower than those over sea during the cruise sampling campaigns. Anthropogenic metallic elements including Zn, Mn, Pb, Cr, and Ni had higher concentrations over sea than those at the offshore islands, suggesting that $PM_{2.5}$ was not only influenced by marine aerosols but also by anthropogenic particles originated from human activities such as industrial processing, fuel burning, and vehicular and shipping exhausts. Higher mass ratios of Ni/Al and Ni/Fe over sea than those at the offshore islands suggested that shipping emissions had higher

influences on marine fine particles than crustal dusts in open sea while compared to those at the offshore islands. The carbonaceous contents of $PM_{2.5}$ indicated that the concentrations of organic carbons (OC) were generally higher than those of elemental carbons (EC). The higher mass ratios of organic and elemental carbons (OC/EC) were observed at the central and north Taiwan Strait, and follow by the offshore islands and the south Taiwan Strait. Overall, sea salts and anthropogenic particles had significant influences on the chemical composition of $PM_{2.5}$ over sea and at the offshore islands.





## 1 Introduction

Sea salts are the most prominent natural emitted particles that might significantly influences regional air quality and global climate change, particularly at the islands and along the coasts. Marine aerosols play an important role in the atmospheric chemistry and physics globally since the ocean occupies about 70% of the earth's surface. Sea salt particles emitted from
oceans/seas could influence the radiative balance of the atmosphere (*Murphy et al., 1998*) and cloud formation (*Pierce and Adams, 2006*). Oceanic sprays are emitted from the surface of oceans/seas by the bursting of white-cap bubbles, while their concentrations are relatively high at the islands and seashores, and descends rapidly with the distance from the coastline (*Chow et al., 1996; Kim et al., 2000; Tsai et al., 2011; Posfai and Buseck, 2010; Seinfeld and Pandis, 2006; Mahowald et al., 2006; Adachi and Buseck, 2015*). Moreover, the concentrations of sea salt particles in the ambient air are also affected by
wind speed, wind direction, elevation, and topography (*Adachi and Buseck, 2015; Hsu et al., 2007*).

Previous studies focused mainly on the sea salt concentrations of segregated aerosol particles (i.e. $PM_{10}$, $PM_{2.5-10}$, and $PM_{2.5}$) and the chemical reactions of gaseous acids with atmospheric particles (e.g. sea salts, crustal dusts, and anthropogenic particles) to form secondary inorganic salts, mainly replacing $Cl^-$ with $SO_4^{2-}$ and $NO_3^-$ and form salts with $Na^+$, $K^+$, $Mg^{2+}$, and $Ca^{2+}$. The spray droplets of marine aerosols regarded as the largest size distribution have a considerable mass content and are
readily deposited on the ground (*Lewandowska et al., 2013*). Anthropogenic sources including industrial processing, fuel and biomass burning, and vehicular and shipping exhausts emitted elsewhere might also influence the chemical composition of marine fine particles *(Tsai et al., 2011; Pandolfi et al. 2011; Pey et al., 2013; Johnson et al., 2014)*. Moreover, the intensity of changes in the chemical composition of marine aerosols is also governed by meteorological factors such as convection, thermal inversion, air humidity, wind speed, and direction, as well as the occurrence of sea and land breezes (*Seinfeld and*
*Pandis 1998; Rastogi and Sarin 2005*).

An offshore island, the Penghu Islands, located at the southeastern Taiwan Strait is approximately 90 km away from the coastline of the Taiwan Island. The Penghu Islands consists of 97 small islands with a population of 108,000 and a total area of 128.0 $km^2$, which has the subtropical weather being mostly influenced by eastern Asian Monsoons. The major air pollution sources in the islands include an oil-fired power plant, a few construction sites, and limited automobiles and fishing
boats. The prevailing winds are blown by the northeastern Monsoons in winter and spring and by the southwestern Monsoons in summer and early fall. Thus, long-range transportation is superior to local sources at the Penghu Islands. Previous studies reported that polluted air masses are mainly transported from the coastal regions of East China, Korean Peninsula, and/or South Japan Islands toward the Penghu Islands (*Yuan et al., 2004*).

Taiwan Strait located between Taiwan and China is one of the busiest marine transportation routes over the worldwide
shipping. Large quantity of anthropogenic air pollutants from stationary and mobile sources could be emitted to the atmosphere (such as particulate pollutants) and transported to and across the Taiwan Strait. Additionally, both sides of the Taiwan Strait are the densely populated and thriving industrial areas, where particulate pollutants could be intensively emitted and deposited over the seas and lands via dry/wet deposition and atmospheric dispersion. Moreover, the impacts of





Asian duststorms, biomass burning, and northeastern Monsoons on ambient air quality are commonly observed in spring and winter in this areas, which leads to a large quantity of Asian dusts and anthropogenic particles transported long-range toward the Taiwan Strait.

The study aims to inquire the chemical composition and spatial variation of chemical characteristic of atmospheric fine

particles and to ascertain how sea salts and anthropogenic particles influence atmospheric $PM_{2.5}$ over sea and at the offshore islands in the Taiwan Strait. The contributions of non-sea salt water-soluble ionic species (nss-WSI) and sea salt water-soluble ionic species (ss-WSI) in the atmospheric $PM_{2.5}$ over sea and at the offshore islands were further investigated in three cruise sampling of marine fine particles during the polluted seasons of winter and spring in 2013-2014.

## 2 Methods

### 2.1 Sampling Protocol of $PM_{2.5}$

Atmospheric $PM_{2.5}$ samples were collected during the cruise sampling campaigns as well as at an offshore island (i.e. the Penghu Islands) located in the southeastern Taiwan Strait. High-volume samplers with cascade impactors were applied to simultaneously collect marine $PM_{2.5}$ in an air quality sampling boat as well as atmospheric $PM_{2.5}$ at the Penghu Islands. Figure 1 illustrates the navigation routes/courses during three cruise sampling campaigns in the Taiwan Strait. In this study,

we weighted the mass concentrations of both $PM_{2.5}$ and $PM_{2.5-10}$ after appropriate conditioning for each sample. However, only $PM_{2.5}$ samples were used for further chemical analysis. $PM_{2.5-10}$ samples cannot be analyzed for chemical composition since they were contaminated with the oil coated on the surface of the impactors. Although sea salts are important component for $PM_{2.5-10}$ samples, this study aimed to investigate the chemical characteristics and spatial variation of marine fine particles ($PM_{2.5}$) in the atmosphere during the highly polluted seasons and how importance o f sea salts and anthropogenic particles

influenced $PM_{2.5}$ in the Taiwan Strait.

During the cruise sampling campaigns, marine $PM_{2.5}$ aerosols were sampled on the board of the air quality sampling boat (R/V OCEAN RESEARCHER III) for three consecutive courses on December $2^{nd}$-$3^{rd}$, 2013 (the winter of 2013, W13), four consecutive courses on April $10^{th}$-$12^{th}$, 2014 (the spring of 2014, S14), and three consecutive courses on December $12^{th}$-$13^{th}$, 2013 (the winter of 2014, W14), respectively. Atmospheric fine particles mixed with marine aerosols were sampled at the

fore of the uppermost deck on the shipboard with a high-volume sampler during the voyage to prevent the interferences from the exhaust gases of the air quality sampling boat itself as well as the seawater sprays emitted from the oceanic surface. The air quality sampling boat sailed to windward during the entire sampling voyage. Consequently, the winds were blown from the prow of the boat in order to avoid the intrusion of oil-burning particles emitted from the air quality sampling boat itself. Each sampling course was arranged to collect $PM_{2.5}$ for continuous 8-12 hours based on the distance of each voyage in the

two- or three-day periods. The courses of $PM_{2.5}$ sampling cruises were named W13C1-W13C3 in the winter cruise of 2013, S14C1-S14C4 in the spring cruise of 2014, W14C1-W14C3 in the winter cruise of 2014. The air quality sampling boat navigated in the Taiwan Strait to collect marine fine aerosols. The objective of cruise sampling over sea in the Taiwan Strait





was to investigate the chemical characteristics and spatial variation of atmospheric marine fine particles during the highly polluted seasons (i.e. spring and winter) in the Taiwan Strait (Li et al., 2013a, 2013b, 2015, 2016, Hsu et al., 2010). However, the differences of three sampling cruises in this study were to inquire the atmospheric marine fine particles at different regions in the Taiwan Strait rather than in different seasons. The navigation routes of the $PM_{2.5}$ sampling cruises were

targeted on southern Taiwan Strait in the winter cruise of 2013, central Taiwan Strait in the spring cruise of 2014, and northern Taiwan Strait in the winter cruise of 2014, respectively.

At the Penghu Islands, also known as the Pescadores Islands, atmospheric $PM_{2.5}$ was sampled on the flat roof of a three-store building at the Xiaomen site ($23^{o}38'47.1"N; 11930'31.6"E$) which is located at the northwestern coastline of the Penghu Islands and is approximately 10 meters above the ground, 60 meters from the seashore, and 1.1 km from the major roads,

respectively.  Xiaomen is a fishery village that is about 35 km far from the downtown of Makung City, the main city of the islands, where is situated at the southeastern coastal area of the Penghu islands, in order to avoid the potential influences from anthropogenic sources such as vehicular exhausts and stationary emissions. The sampling of $PM_{2.5}$ was conducted for continuous 24 hours (from 9:00 am to 9:00 am of the sequential day) in the seven-day periods on December $1^{st}$-$8^{th}$, 2013, April $8^{th}$-$15^{th}$, 2014, and December $9^{th}$-$16^{th}$, 2014, respectively. Same high-volume sampler with a cascade impactor was also

applied to collect $PM_{2.5}$ samples at the Xiaomen site.

## 2.2 Chemical Analysis of $PM_{2.5}$

After sampling $PM_{2.5}$, the quartz fiber filters were temporarily stored at $4^{\circ}C$ to conserve their chemical stability, and then transported back to the Air Pollution Laboratory of the Institute of Environmental Engineering at National Sun Yat-sen University within two days for conditioning, weighing, and further chemical analysis. All quartz fiber filters were divided

into four identical parts prior to the chemical analysis. One quarter of each quartz fiber filter was analyzed for water-soluble ionic species of $PM_{2.5}$ by means of ion chromatography (IC) (Dionex, DX-120). The concentrations of the major anions ($F^{-}$, $Cl^{-}$, $SO_4^{2-}$, and $NO_3^{-}$) and cations ($NH_4^{+}$, $K^{+}$, $Na^{+}$, $Ca^{2+}$, and $Mg^{2+}$) were measured. The quartz fiber filters analyzed for ionic species were put into a 15-ml PE bottle, and distilled de-ionized water (D.I. $H_2O$) was added into each PE bottle for ultrasonic vibration of 60 min or longer.

Another quarter of each quartz fiber filter was digested by microwave digestion method in a 30 mL mixed acidic solution ($HNO_3$:HCl=1:3) by heating it up to 150-200°C for 2 hours, and then diluted to 50 ml with distilled de-ionized water (D.I. $H_2O$) for further analysis of metallic elements. Twelve metallic elements of $PM_{2.5}$ including Na, Ca, Al, Fe, Mg, K, Zn, Cr, Ti, Mn, Ni, and Pb were analyzed with an inductively coupled plasma-atomic emission spectrometer (ICP-AES) (Perkin Elmer, Optima 2000DV).

The carbonaceous contents (i.e. elemental, organic, and total carbons) of $PM_{2.5}$ were measured with an elemental analyzer (EA) (Carlo Erba, Model 1108). Prior to sampling, the quartz fiber filters had to be pre-heated at 900 °C for 1.5 h to expel the potential organic impurities. This preheating procedure minimized the background carbonaceous species in the quartz fiber filters and matrix, which would interfere with the analytical results, possibly leading to an overestimation of the





carbonaceous species of $PM_{2.5}$. The elemental analyzer was operated using the procedure of oxidation at 1020°C and that of reduction at 500°C, for continuous 15-min heating. Additionally, one eighth of the quartz fiber filter was heated in advance by hot nitrogen gas (340-345°C) for 30 min to expel the organic carbon (OC) fraction, after which the amount of elemental carbon (EC) was determined. Another eighth of the quartz fiber filter was analyzed without heating, and the carbonaceous

species thus characterized as total carbon (TC). The amount of organic carbon (OC) could be determined by subtracting the elemental carbon from total carbon. Although the aforementioned thermal analysis was the most widely used method for determining the carbonaceous species in marine $PM_{2.5}$ aerosols, a charring formation error from filter preheating was not taken into account for correction, and this artifact might cause the overestimation of EC and the underestimation of OC (*Li et al., 2013a; 2013b ;2015; 2016*).

**2.3 Quality Assurance and Quality Control**

Quality assurance and quality control (QA/QC) for both sampling and chemical analysis of $PM_{2.5}$ were also employed in this study. Prior to sampling, the flow rate of each $PM_{2.5}$ sampler was carefully calibrated with an orifice calibrator (SENSIDYNE, MCH-01) (Xcalibrator high-volume air sampler calibrator TE-HVC-101). The quartz fiber filters were handled with care, so as to prevent potential contamination and cracking during the sampling procedure, as they were placed

on the $PM_{2.5}$ samplers. After sampling, aluminium foil was used to fold the quartz fiber filters, which were then temporarily stored at 4°C and transported back to the laboratory for chemical analysis within two days. The sampling and analytical procedures were similar to those described in previous studies (*Cheng and Tsai, 2000; Lin, 2002; Yuan et al., 2006; Tsai et al., 2008; Tsai et al., 2010; Witz et al., 1990*).

Both field and transportation blanks were further undertaken for $PM_{2.5}$ sampling, while reagent and filter blanks were applied

for chemical analysis. The determination coefficient ($R^2$) of the calibration curve for each chemical species was required to be higher than 0.995. Background contamination was routinely monitored by using operational blanks (unexposed filters); these were processed simultaneously with the field samples. The present experimental works found that the background interference was insignificant and could thus be ignored in this study. At least 10% of the samples were analyzed by spiking with a known amount of metallic elements and water-soluble ionic species to determine their recovery efficiencies. The

results of the recovery efficiency tests indicated that the recovery efficiencies among every 10 filter samples varied from 96 to 103%. In addition, the results of the reproducibility varied from 97 to 104% for all the chemical species.

**3 Results and discussion**

**3.1 Spatiotemporal Variation of $PM_{2.5}$ Concentrations over Sea and at the Offshore Islands**

The mass concentrations of $PM_{2.5}$ and $PM_{2.5-10}$, and the mass ratios of $PM_{2.5}/PM_{10}$ over sea and at the offshore islands during

the $PM_{2.5}$ sampling campaigns are summarized in Table 1. Limited to the voyage of air quality monitoring boat, each





sampling course was arranged to collect $PM_{2.5}$ for only continuous 8-12 hours (i.e. daytime and nighttime) based on the distance of navigation during the sampling cruise over sea. At the islands, $PM_{2.5}$ was sampled for continuous 24 hours.

Field sampling results indicated that the concentrations of $PM_{2.5}$ were generally higher than those of $PM_{2.5-10}$ with the exception of the winter cruise in 2014 (W14).

The mass ratios of $PM_{2.5}/PM_{10}$ over sea and at the offshore islands ranged from 42.4% to 44.0% for the winter cruise of 2014 (W14), which were lower than those for other two sampling cruises ranging from 49.2% to 55.7%. The concentrations of $PM_{2.5-10}$ over sea were always higher than those at the offshore islands, indicating that sea salts highly influenced the marine aerosols with a dominant size range of 2.5-10 μm. Higher $PM_{2.5}$ concentrations were commonly observed over sea. The fine particles might be emitted from vessels including commercial ships and fishery boats were significant contributions to

marine fine aerosols and cannot be ignored over the seas and oceans where the marine transportation traffics are busy. Unexpectedly, a similar trend was also observed for $PM_{2.5}$.with an exception of winter cruise of 2013 (W13). There are two possible suggestions or explanations for this phenomenon. One possibility suggested that fine particles emitted from vessels including commercial ships and fishery boats were significant contributions to marine fine aerosols and cannot be ignored over the seas and oceans where the marine transportation traffics are busy. The other possibility indicated that long-range

cross-boundary transportation of fine particles emitted from the upwind regions such as China, Korea, and/or Japan could be constantly transported to the target areas and even across the Taiwan Strait.

Moreover, higher $PM_{2.5}$ concentrations were commonly observed in the daytime than those at nighttime for all three cruise sampling campaigns. It was mainly attributed to the facts that higher frequency of human activities such as commercial burning, vehicle and vessel traffics in the daytime than at nighttime. Additionally, the sea-land breezes play an important

role in transporting air pollutants to and from urban areas on the coastal regions *(Ding et al., 2004; Tsai et al., 2008)*. Previous study stated that the surface wind fields at the Penghu Islands varied in the daytime and at nighttime (Li et al., 2015). In the daytime, air masses were divided into two separate zones. On the west side of the Penghu Islands, a strong northward prevailing wind was coupled with the sea-land breeze, causing by the sea breezes in the southwestern coastal region of the Taiwan Island. Conversely, the surface wind directions moved to the east on the east-side of the Penghu Islands.

*Viana et al. (2005)* also stated that atmospheric coastal dynamics exert a significant influence on the levels and composition of atmospheric particles. Sea breezes may penetrate deep inland and cause high pollutant episodes in early afternoon, resulting in higher concentration of marine aerosols in the daytime than that at nighttime *(Ding et al., 2004)*.

**3.2 Water-soluble Ions of $PM_{2.5}$ over Sea and at the Offshore Islands**

The water-soluble ionic species of $PM_{2.5}$ sampled over sea and at the offshore islands in the Taiwan Strait during the cruise sampling campaigns are summarized in Table 2. The most abundant water-soluble ionic species of $PM_{2.5}$ were $NO_3^-$, $SO_4^{2-}$, $NH_4^+$, $Cl^-$, and $Na^+$ indicating that secondary inorganic aerosols (SIAs) and sea salts were the major portion of $PM_{2.5}$. The most possible inorganic compounds of $PM_{2.5}$ were ammonium nitrate ($NH_4NO_3$), ammonium sulfate (($NH_4)_2SO_4$), which





were formed by the neutralization of sulfuric and nitric acids with ammonia. The results indicated that both sea salts and anthropogenic secondary particles were the major contributors to marine fine aerosols in the Taiwan Strait (*Li et al., 2016*). Particularly, the mass concentrations of $SO_4^{2-}$ (3.6-6.5 µg m$^{-3}$) were approximately 1.4-1.9 times higher than those of $NO_3^-$ (2.5-4.0 µg m$^{-3}$), indicating that anthropogenic particles came mainly from the sulfur-containing sources rather than the

nitrogen-containing sources.

Moreover, the contributions of sea salts (Cl$^-$ and Na$^+$) to PM$_{2.5}$ over sea were commonly higher than those at the offshore islands. It was mainly attributed to the facts that sea salt particles could be massively generated from the surface of the ocean/sea by the bursting of white-cap bubbles (*Chow et al., 1996*). The concentrations of sea salts in marine aerosols are the highest over sea and then descend rapidly with the distance away from the coastline (*Kim et al., 2000; Tsai et al., 2011*). The

replacement of chloride from sea-salt particles with acids is caused by the formation of sulfate and nitrate in the marine aerosols. Particularly, nitric acid prefers to react with sodium chloride (NaCl) to form stable sodium nitrate (NaNO$_3$) over sea and at the offshore islands (*Wall et al., 1988; Fang et al., 2000*).

The percentages of SIAs (i.e. the summation of $SO_4^{2-}$, $NO_3^-$, and $NH_4^+$) ranged from 13.4% to 16.9%, from 8.8% to 11.5%, and from 3.9% to 6.7% in PM$_{2.5}$ collected in the courses of W13C1, S14C1, and W14C3, respectively, that was navigated in

the southern Taiwan Strait. In these courses, the collected fine particles were mainly considered as a mixture of marine aerosols and anthropogenic particles emitted from southern Taiwan, particularly the Kaohsiung City.  The contributions of $SO_4^{2-}$, $NO_3^-$, Cl$^-$, and Na$^+$ to PM$_{2.5}$ collected in the southern Taiwan Strait were obviously higher than other regions. The accumulation of particulate matter in the near-ocean region due to sea-land breeze had a regular influence on the physicochemical properties of atmospheric particles in the coastal region of southern Taiwan (*Tsai et al., 2010*). The air

masses for PM$_{2.5}$ samples W13C2, S14C2, and W14C2 were mainly from marine environments adjacent to the Penghu Islands. The concentrations of water-soluble ionic species in PM$_{2.5}$ for these courses were relatively lower than those for other courses. Moreover, higher contributions of SIAs to PM$_{2.5}$ were generally observed in the daytime than those at nighttime in the southern Taiwan Strait. The second highest contributions of water-soluble ionic species to PM$_{2.5}$ were observed in the courses of W13C3, S14C4, and W14C3, which navigated along the coast of the Taiwan Island on its way

back to the Kaohsiung Harbor. In these three courses, PM$_{2.5}$ was considered as a mixture of marine aerosols and anthropogenic particles mainly emitted from pollution sources in central Taiwan Island where the largest coal-fired power plant in Taiwan is located at the western coast of Taiwan Island.

### 3.3 Contributions of Sea Salts to PM$_{2.5}$ over Sea and at the Offshore Islands

The concentrations of sea salts can be estimated by the following three equations (equations (1)-(3)) cited in the previous

literature as shown below, where 1.47 is the mass ratio of (Na$^+$+K$^+$+Mg$^{2+}$+Ca$^{2+}$+SO$_4^{2-}$+ HCO$_3^-$)/Na$^+$.

Sea salt = Cl$^-$+1.47×Na$^+$ (1)

Sea salts = Cl$^-$+Na$^+$ (2)

Sea salts = ss-K$^+$+ss-SO$_4^{2-}$+ss-Ca$^{2+}$+ss-Mg$^{2+}$+ss-NO$_3^-$+Cl$^-$+Na$^+$ (3)





Table **2** summarizes the estimated mass concentrations of sea salts and their contributions to $PM_{2.5}$ over sea and at the offshore islands during three cruise sampling campaigns. Although the mass concentrations of sea salts estimated by three equations were similar, the sea salt concentrations estimated by equation (3) were always higher than those estimated by equations (1) and (2). Previous studies reported that the mass of sea salts can be adopted by the sum of $Cl^-$ and $Na^+$ (equation

(2)) (*Chow et al., 1996; Kim et al., 2000; Tsai et al., 2011; Han et al., 2003; Virkkula et al., 2006*). Besides, the mass ratios of various water-soluble ionic species ($Na^++K^++Mg^{2+}+Ca^{2+}+SO_4^{2-}+ HCO_3^-$) to sodium ($Na^+$) in seawater have distinction in different investigated region. However, this study revealed that the hypothesis of all $Na^+$ derived from seawater would be overestimated by the equation (3). Inter-comparing with these three estimated equations, this study found that the most accurate method to estimate the sea salt concentrations was equation (1).

As shown in Table 2, the estimated sea salt results indicated that the concentrations of sea salts over sea (2.9-6.5 µg m$^{-3}$) were always higher than those at the offshore islands (2.4-4.0 µg m$^{-3}$), while the contributions of sea salts to $PM_{2.5}$ over sea (10.5-18.1%) were constantly higher than those at the offshore islands (6.5-14.7%) during the cruise sampling campaigns. Previous studies reported that the sea salt concentrations in $PM_{10}$ was 53.4% at Point Reyes, Central CA (*Chow et al., 1996*), 24.7% at San Nicolas Island in South Coast Air Basin of Southern California (*Kim et al., 2000*), 11.2~15.8% at the coastal

region of Southern Taiwan (*Tsai et al., 2011*), 3.1~13.5% at Gosan, Jeju, South Korea (*Han et al., 2003*), and 2.81% and 1.62% in $PM_{10}$ and $PM_{2.5}$, respectively, at the coastal region of South Korea (*Park et al., 2015*), respectively. The contributions of sea salts to $PM_{2.5}$ were consistent well with *Tsai et al. (2011)* in the South Taiwan and *Han et al. (2003)* in South Korea.

**3.4 Comparison of Sea Salt and Non-Sea Salt Content of PM$_{2.5}$**

Sea salts are one of the abundant aerosols in the atmosphere of the coastal regions (*Posfai and Buseck, 2010; Seinfeld and Pandis, 2006; Mahowald et al., 2006; Adachi and Buseck, 2015*). Previous studies reported that sodium ($Na^+$) is an excellent tracer of sea salts elsewhere, particularly in the coastal regions (*Ooki et al., 2002*). Other alternative tracers of sea salts would be $Cl^-$ or $Mg^{2+}$ which are abundant in the seawater. In the atmosphere, chloride ($Cl^-$) of the aerosol particles along the coastline or even intrusion to far inland could be partially depleted due to the chemical reactions of sea salts with acidic

compounds such as sulfuric or nitric acids. This approach prevents the inclusion of non-sea salt ions such as $K^+$, $Mg^{2+}$, $Ca^{2+}$, $SO_4^{2-}$, and $NO_3^-$ in the sea salt particles and allows for the loss of $Cl^-$ through chloride depletion process, assuming that all measured $Na^+$ in aerosol particles is derived from seawater.

We further reviewed the literatures for the mass ratios of various water-soluble ions ($K^+$, $Mg^{2+}$, $Ca^{2+}$, $SO_4^{2-}$) to sodium ($Na^+$) in the seawater. Previous studies have commonly used these ratios to estimate the non-sea-salts water-soluble ions in the

aerosol particles. The non-sea salt water-soluble $K^+$, $Mg^{2+}$, $Ca^{2+}$, $SO_4^{2-}$, and $NO_3^-$ can be estimated by equations (4)-(8).

$$nss\text{-}K^+ = K^+-0.038\times Na^+ \tag{4}$$

$$nss\text{-}Mg^{2+} = Mg^{2+}-0.12\times Na^+ \tag{5}$$

$$nss\text{-}Ca^{2+} = Ca^{2+}-0.038\times Na^+ \tag{6}$$





nss-SO$_4^{2-}$ = SO$_4^{2-}$-0.251×Na$^+$ (7)

nss-NO$_3^-$ = [NO$_3^-$]-ss-[NO$_3^-$] = [NO$_3^-$]-(1.174[Na$^+$]-[Cl$^-$]) (8)

Figure 2 and Table 3 show the water-soluble ion concentrations and the contributions of sea salt and non-sea salt water-soluble ion (ss- and nss-WSI) concentrations in PM$_{2.5}$ over sea and at the offshore islands in the winter and spring of 2013-

5 2014. The major nss-WSI in PM$_{2.5}$ were non-sea salt SO$_4^{2-}$ (nss-SO$_4^{2-}$) and non-sea salt NO$_3^-$ (nss-NO$_3^-$), while the major ss-WSI in PM$_{2.5}$ were ss-Cl$^-$ and ss-Na$^+$, over sea and at the offshore islands. The percentages of nss-WSI in PM$_{2.5}$ were ordered as: nss-SO$_4^{2-}$ (10.8%-15.0%) > nss-NO$_3^-$ (5.4%-10.3%) > NH$_4^+$ (3.7%-6.7%) > nss-K$^+$ (0.7%-2.2%) > nss-Mg$^{2+}$ (1.0%-2.2%) > nss-Ca$^{2+}$ (0.9%-1.3%), while the percentages of ss-WSI in PM$_{2.5}$ were in order of Cl$^-$ (6.3-9.7%) > Na$^+$ (4.0-6.0%) > ss-NO$_3^-$ (0.6-2.4%) > ss-SO$_4^{2-}$ (1.0-1.5%) > ss-Mg$^{2+}$ (0.5-0.7%) > ss-K$^+$ (0.2-0.2%) > ss-Ca$^{2+}$ (0.2-0.2%), respectively. These

10 results indicated that the concentrations of nss-WSI (6.4-11.8 μg m$^{-3}$) in PM$_{2.5}$ were much higher than those of ss-WSI (3.3-6.5 μg m$^{-3}$) both over sea and at the offshore islands. The results indicated that atmospheric PM$_{2.5}$ in the Taiwan Strait was mainly influenced by anthropogenic particles rather than sea salts. This phenomenon was not only observed at the offshore islands, but was also found in the open sea. Furthermore, the contributions of nss-WSI to PM$_{2.5}$ at the offshore islands were obviously higher than those over sea, while the contributions of ss-WSI to PM$_{2.5}$ at the offshore islands were generally lower

than those over sea, respectively. It suggested that anthropogenic particles emitted from inland sources or local emissions could be probably accumulated at the offshore islands rather than over sea.

Figure 3 illustrates the percentages of non-sea salt water-soluble ionic species in the measured water-soluble ionic species of PM$_{2.5}$ sampled over sea and at the offshore islands during the cruise sampling campaigns. The contribution of nss-SO$_4^{2-}$ to total SO$_4^{2-}$ ranged from 90.8% to 92.9% over sea and from 93.7% to 96.2% at the offshore islands, respectively. Similar

trends were observed for the mass percentages of nss-K$^+$ to total K$^+$ (96.0-96.9%) as well as nss-Ca$^{2+}$ to total Ca$^{2+}$ (83.1-93.2%) at the offshore islands. Previous study reported that water-soluble ionic species of Na$^+$ and Mg$^{2+}$ are usually expected to be conservative tracers of sea salts, while Ca$^{2+}$ may have additional sources, such as continentally derived gypsum or oceanic CaCO$_3$ (*Johansen et al., 1999*). In this study, the remaining nss-Ca$^{2+}$ was assumed to be present as CaCO$_3$.3.5 Ratio of Chloride to Sodium Ions and Chloride Deficit of PM$_{2.5}$

Previous studies reported that sea salt particles contribute significant fraction to particulate matter at the locations close to the sea (*Chow et al., 1996; Manders et al., 2009; Tsai et al., 2011; Park et al., 2015*). The ratio of Cl$^-$/Na$^+$ in fresh sea salts is about 1.8 in the mass fraction (*Pytkowicz and Kester, 1971*) or 1.16 (*Riley and Chester, 1971*) in the ionic equivalents. The chloride (Cl$^-$) in the sea salt particles could be continuously lost to the gas phase as HCl and chlorine compounds while acidic species existed in the atmosphere. Chloride deficit is the process by which acidic species, mainly nitrate, sulfate, and

organic acids, react with NaCl in the sea salt particles and replace chloride in the form of HCl. *Virkkula et al. (2006)* reported that in the polluted areas where strong acids (H$_2$SO$_4$ and HNO$_3$) or their precursors are present, simple acid displacement reactions are likely responsible for most of the chloride depletion. These reactions lead to large (even up to 100%) chloride deficits if enough acidic species and/or time are available. The chloride deficit can be calculated by equation (9) as follows (*Quinn et al., 2000*),





$$Cl^- \; deficit \; (\%) = \frac{[Cl^-]_{original} - [Cl^-]_{meas}}{[Cl^-]_{original}} \times 100\% = \frac{1.8 \times [Na^+]_{meas} - [Cl^-]_{meas}}{1.8 \times [Na^+]_{meas}} \times 100\% \tag{9}$$

where $1.8 \times [Na^+]$ is the expected $Cl^-$ concentration in the original sea salts, in the absence of any loss of $Cl^-$ and all $Na^+$ in the marine aerosols of sea-salt origin, $Cl^-_{meas}$ is the chloride measured in the aerosol particles.

Previous studies indicated that the mass ratio of $Cl^-/Na^+$ has a tendency to decrease as the distance from the sea or the coast increases due to the depletion of $Cl^-$ (*Chow et al., 1996; Kim et al., 2000; Dasgupta et al., 2007; Park et al., 2015*). Figure 4 illustrates the chloride deficit and the molar ratio of $Cl^-/Na^+$ in $PM_{2.5}$ over sea and at the offshore islands in the Taiwan Strait. The molar ratio of $Cl^-/Na^+$ ranged from 0.93 to 1.08 over sea and from 0.94 to 0.98 at the offshore islands, respectively. *Park et al. (2015)* reported that the molar ratio of $Cl^-/Na^+$ varies significantly with locations, generally high at the coastal sites and low at the inland and urban sites. However, this cruise campaign showed no significant trend of chloride deficit over sea and at the offshore islands. Narrow chloride deficits ranging from 16.2% to 19.2% were observed at the offshore islands, while the chloride deficit varied significantly from 7.6% to 23.0% over sea. Lower chloride deficits were mostly observed for the cruise courses navigated along the coastline of Taiwan Island during the cruise sampling campaigns.

The chloride deficit measured in the marine aerosols sampled at the coastal regions and open seas are compared and summarized in Table 4. In East China Sea, the $Cl^-$ deficit reported by *Park et al. (2004)* was accounted for 55% for fine-mode particles and 16% for coarse-mode particles during the normal periods. However, *Hsu et al. (2007)* reported that the $Cl^-$ deficits were higher 86% and 98% for fine-mode particles; and 29% and 30% for coarse-mode particles sampled at the Dongsha islands in South China Sea. The higher $Cl^-$ deficit on South China Sea was obviously higher than those on East China Sea. In this study, the $Cl^-$ deficit at the offshore site and over sea in the Taiwan Strait ranged from 16.2% to 19.2% at the offshore islands and from 7.6% to 20.4% over sea, respectively. Previous study indicated that the aged nature of sea salt particles were about 150 km from the open sea, giving these particles enough time to react with atmospheric acidic gases (*Virkkula et al., 2006*). The East China Sea and the Taiwan Strait received much more Asian outflow of acidic gases from Asian continent (*Streets et al., 2000; Carmichael et al., 2002; Uematsu, et al., 2010*), while air masses transported toward South China Sea were not only blown from South China but also from biomass burning emitted from the Southeast Asia, particularly the Indochina Peninsula (*Arndt et al., 1997*). The aged air masses were transported toward the South China Sea, causing higher $SO_2$-rich air pollutants over sea. According to *Li et al. (2016)*, their results indicated that the averaged $Cl^-$ deficits ranged from 38.2% to 43.4% at the west-side sites, from 37.0% to 42.7% at the east-side sites, and from 16.2% to 19.4% at the offshore site, respectively, in the Taiwan Strait. The results of $Cl^-$ deficits obtained from these cruise sampling campaigns were consistent quite well with *Park et al. (2004) and Tsai et al. (2010)*, which showed a tendency to increase with the distance from the coastline.

### 3.6 Metallic Elements of $PM_{2.5}$ over Sea and at the Offshore Islands

Previous studies reported that the metallic elements of $PM_{2.5}$ are probably attributed from three major sources, including crustal, anthropogenic, and oceanic sources (*Chester et al., 2000; Zhao et al., 2015*). The metallic elements lead (Pb) and



zinc (Zn) are often considered as the markers associated with traffic sources (*Querol et al., 2001; Manoli et al., 2002; Fang et al., 2003; Cao et al., 2009*). Potassium (K) is associated with primary particles emitted from coal combustion and biomass burning. Zn, Ni, Pb, and Mn were largely from the industrial emissions such as metallurgical processes (*Pacyna, 1998*), while Ni and Zn could be also partially from the tail exhausts of motor vehicles and vessels (*Lee et al., 1999*). *Zhao et al.*

*(2015)* reported that atmospheric pathway has been recognized as an important source for many oceanic chemicals, including Fe (*Duce and Tindale, 1991; Gao et al., 2001*), Al (*Kang et al., 2009*), Cu (*Kocak et al., 2005*), Zn (*Spokes et al., 2001; Kang et al., 2009*), and Pb (*Lin et al., 2000; Kocak et al., 2005; Kang et al., 2009*). Many studies concluded that the concentrations of crustal-originated (Al, Ca, and etc.) atmospheric particles were obviously decreased when air masses were blown from their origins to the downwind regions during the long-range transport (LRT) processes (*Gao et al., 1992a;*

*Spokes et al., 2001; Han et al., 2008; Zhao et al., 2015*).

Table 5 shows that the most abundant metallic elements of $PM_{2.5}$ were crustal elements (Al, Fe, Mg, K, and Ca), and followed by anthropogenic elements (Zn, Ni, and Pb). The highest metallic element concentrations of $PM_{2.5}$ were commonly observed at the offshore islands, while compared to those observed over sea. Among the crustal elements, Al, Fe, and Ca were the richest metals, with a much higher concentration of Al than other crustal elements. Metallic element Al is often

considered as an indicator of mineral dusts (*Hsu et al., 2010*), which is one of the major common crustal-originated elements in the dust particles. *Kang et al. (2011)* reported that high Al concentration is generally observed during the Asian duststorm events in East Japan Sea. The percentages of Al to $PM_{2.5}$ account for 12.4% in Yellow Sea and East China Sea (*Zhao et al., 2015*), which are higher than those in Bohai Sea (*Ji et al., 2011*). In this study, the percentages of Al to $PM_{2.5}$ ranged from 2.8% to 3.2% at the offshore islands and from 1.5% to 2.2% over sea, respectively. The percentages of Al to $PM_{2.5}$ in the

Taiwan Strait were lower than those in Yellow Sea and East China Sea since the Taiwan Strait were located at the lower latitude regions than those in Yellow Sea and East China Sea regions, where air masses containing dusts could be blown from the dust-originated sources to the downwind regions. The percentages of metallic elements of Mg and Ca to $PM_{2.5}$ ranged from 1.7% to 2.9% and from 1.5% to 2.8% at the offshore islands, respectively, while those ranged from 2.1% to 3.2% and from 1.5% to 3.0% over sea in the Taiwan Strait, respectively. The metallic elements of Mg and Ca over sea were

higher than those at the offshore site. It was probably attributed to the coupling of seawater and terrestrial substances, and some metallic element of Ca might be originated from anthropogenic sources (*Zhang et al., 2007; Zhao et al., 2015*). The percentages of K to $PM_{2.5}$ ranged from 1.8% to 2.5% at the offshore islands and from 1.8 % to 3.3% over sea in the Taiwan Strait, respectively. The metallic element of K in $PM_{2.5}$ was generally originated from biomass burning and marine aerosols (*Chow et al., 1995*). Higher contributions of K in $PM_{2.5}$ were generally observed over sea than those at the offshore islands in

the Taiwan Strait, suggesting that the bubbles of sea salts in the marine environments could emit and transport K-rich particles in the atmosphere.

Nickel (Ni) as an anthropogenic metal has been reported as one of two recognized indicators for shipping emissions (*Pandolfi et al. 2011; Pey et al., 2013; Johnson et al., 2014*), while Al and Fe are two major tracers for crustal emitted dusts (*Hsu et al., 2010*). Consequently, the mass ratios of Ni/Al and Ni/Fe could be treated as important indicators of particles





emitted from vessel exhausts. As shown in **Table 5**, the mass ratios of Ni/Al and Ni/Fe over sea (0.35 and 0.88, respectively) were higher than those at the offshore islands (0.30 and 0.69, respectively). The results indicated that shipping emissions had higher influences on the marine fine particles than crustal dusts in the open sea while compared to those at the offshore islands, which concurred with previous study reported by *Li et al. (2016)*.

5 **3.7 Carbonaceous Contents of PM$_{2.5}$ over Sea and at the Offshore Islands**

Previous studies indicated that the mass ratio of OC and EC (OC/EC) has been widely used in previous studies to determine whether the carbonaceous aerosol particles are mainly primary or secondary (*Ho et al., 2015; Li et al., 2016*). A high OC/EC ratio coupled with a poor correlation implies an influx of urban air pollutants from elsewhere or the formation of secondary organic carbon (SOC) from photochemical reactions (*Castro et al., 1999*). On the other hand, a high correlation indicates 10 primary emission and a secondary formation derived from the primary organic carbon (POC) (*Turpin et al., 1991; Strader et al., 1999; Li et al., 2016*).

The concentrations of SOC can be estimated by using an EC-tracer method (*Khan et al., 2016*). In most cases of previous studies, the primary OC to EC ratio, (OC/EC)$_{pri}$, could be simplified as the minimum OC to EC ratio, (OC/EC)$_{min}$. If the primary OC/EC ratio is available, we can then determine the SOC and POC concentration of PM$_{2.5}$ by equations (10) and 15 (11).

$$SOC = OC\text{-}EC \times (OC/EC)_{pri} \tag{10}$$

$$POC = OC\text{-}SOC \tag{11}$$

However, the (OC/EC)$_{pri}$ is a source- and seasonal-specific parameter, and is also affected by the carbonaceous species determination method (*Khan et al., 2016*). The observed minimum OC/EC ratio, (OC/EC)$_{min}$, at a specific sampling site was 20 often used to represent the (OC/EC)$_{pri}$, assuming that the meteorological conditions are not favorable for the formation of SOC during the sampling periods (*Turpin et al., 1995*).

Figure 5 illustrates the concentrations of estimated secondary OC (SOC), primary OC (POC), and elemental carbon (EC) in PM$_{2.5}$ sampled over sea and at the offshore islands in the Taiwan Strait. The SOC and POC concentrations of PM$_{2.5}$ are estimated by equations (10) and (11). This study estimated the primary OC/EC ratio ((OC/EC)$_{pri}$) as the minimum ratio of 25 particulate OC to EC at the offshore islands in the winter and spring of 2013-2014, representing the primary OC and EC of PM$_{2.5}$.

The concentrations of POC, SOC, and EC in PM$_{2.5}$ and the mass ratios of OC/EC over sea and at the offshore islands during the cruise sampling campaigns are illustrated in Figure 5. The results indicated that the concentrations of OC were always higher than those of EC in PM$_{2.5}$. Moreover, the concentrations of primary OC were always higher than those of secondary 30 OC both over sea and at the offshore islands. Previous studies reported that primary OC are emitted directly from combustion sources, while secondary OC is mainly produced by gas-to-particle conversion or chemical reactions (*Cao et al., 2005; Zhang et al., 2015*). This study revealed that the primary OC was major source of PM$_{2.5}$ over sea and at the offshore islands while compared to the secondary OC.



**4 Conclusions**

The mass concentrations and chemical characteristics of marine fine aerosols over sea at the offshore islands during cruise sampling campaigns in the Taiwan Strait were investigated. Overall, the concentrations of $PM_{2.5}$ were generally higher than those of $PM_{2.5-10}$ in the Taiwan Strait. The concentrations of $PM_{2.5-10}$ over sea were always higher than those at the offshore

islands, indicating that sea salts highly influenced the marine aerosols. A similar trend was also observed for $PM_{2.5}$, suggesting that fine particles emitted from vessel exhausts were significant contributions to marine fine aerosols and cannot be ignored in the Taiwan Strait. Another possibility was that anthropogenic fine particles emitted from upwind regions such as China, Korea, and/or Japan could be transported toward the Taiwan Strait. Moreover, higher $PM_{2.5}$ concentrations were commonly observed in the daytime rather than those at nighttime.

Chemical analysis of $PM_{2.5}$ indicated that both secondary inorganic aerosols (SIAs) and sea salts were the major portion of $PM_{2.5}$ in the Taiwan Strait. The mass concentrations of $SO_4^{2-}$ were higher than those of $NO_3^-$, implying that anthropogenic fine particles in the Taiwan Strait came mainly from the sulfur-originated sources rather than the nitrogen- originated sources. The contributions of sea salts to $PM_{2.5}$ sampled over sea were higher than those at the offshore islands. Moreover, the mass concentrations of estimated nss-WSI in $PM_{2.5}$ were much higher than those of ss-WSI both at the offshore islands and over

sea, indicating that fine particles in the Taiwan Strait was mainly influenced by anthropogenic particles rather than sea salts. This phenomenon was not only observed at the offshore islands, but was also found in the open sea. The contributions of nss-WSI to $PM_{2.5}$ at the offshore islands were obviously higher than those over sea, while the contributions of ss-WSI to $PM_{2.5}$ at the offshore islands were generally lower than those over sea.

The major metallic contents of marine $PM_{2.5}$ were crustal elements and followed by anthropogenic elements. The

20 concentrations of anthropogenic metallic elements (Zn, Mn, Pb, Cr, and Ni) over sea were higher than those at the offshore islands. These anthropogenic metallic elements came probably from industrial process emissions, fuel and coal burning, and vehicular and vessel exhausts. However, higher mass ratios of Ni/Al and Ni/Fe over sea (0.35 and 0.88, respectively) than those at the offshore islands (0.30 and 0.69, respectively) suggested that shipping emissions had higher influences on marine fine particles than crustal dusts in open sea while compared to those at the offshore islands. This study also revealed that

primary organic carbons (POC) were major source of $PM_{2.5}$ over sea and at the offshore islands while compared to secondary organic carbons (SOC).

This cruise sampling campaign further showed no significant trend of chloride deficit over sea and at the offshore islands. A narrow chloride deficit ranging from 16.2% to 19.2% was observed at the offshore islands, while the chloride deficits varied significantly from 7.6% to 23.0% over sea. Lower chloride deficits were commonly observed for the cruise courses

navigated along the coastline of Taiwan Island during the cruise sampling campaigns.





**Acknowledgments**

This study was performed under the auspices of National Sun Yat-sen University and National Science Council of the ROC (Taiwan) for financial support under the Project No. of NSC 102-2628-M-241-001-007.

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

30



**Table 1.** The mass concentrations of $PM_{2.5}$ and $PM_{2.5-10}$, and the mass ratios of $PM_{2.5}/PM_{10}$ over sea and at the offshore islands during three cruise sampling campaigns.

| Sampling Sites | Sampling Abbr. | Day/Night Samples | Sampling Duration | Sampling Periods | $PM_{2.5}$ ($\mu g\ m^{-3}$) | $PM_{2.5-10}$ ($\mu g\ m^{-3}$) | $PM_{10}$ ($\mu g\ m^{-3}$) | $PM_{2.5}/PM_{10}$ (%) | Reference |
|---|---|---|---|---|---|---|---|---|---|
| Islands | W13 | Daily | 24 hrs | Winter 2013 | 28.2 | 23.3 | 51.5 | 54.8 | |
| | S14 | Daily | 24 hrs | Spring 2014 | 22.7 | 19.7 | 42.4 | 53.5 | |
| | W14 | Daily | 24 hrs | Winter 2014 | 38.1 | 30.3 | 68.3 | 55.7 | |
| Over Sea | W13C1 | Nighttime | 8 hrs | Winter 2013 | 21.5 | 22.2 | 43.7 | 49.2 | This study |
| | W13C2 | Daytime | 12 hrs | | 27.0 | 22.2 | 49.2 | 55.0 | |
| | W13C3 | Nighttime | 12 hrs | | 26.0 | 23.4 | 49.3 | 52.6 | |
| | S14C1 | Nighttime | 12 hrs | Spring 2014 | 30.2 | 29.3 | 59.5 | 50.8 | |
| | S14C2 | Daytime | 12 hrs | | 31.7 | 31.1 | 62.7 | 50.5 | |
| | S14C3 | Nighttime | 12 hrs | | 31.3 | 30.5 | 61.8 | 50.7 | |
| | S14C4 | Daytime | 16 hrs | | 33.4 | 31.2 | 64.6 | 51.7 | |
| | W14C1 | Nighttime | 8 hrs | Winter 2014 | 38.6 | 52.5 | 91.1 | 42.4 | |
| | W14C2 | Daytime | 12 hrs | | 43.5 | 55.3 | 98.8 | 44.0 | |
| | W14C3 | Nighttime | 12 hrs | | 38.3 | 51.0 | 89.3 | 42.9 | |
| West-side of the Taiwan Strait | XM,FZ | Daily | 24 hrs | Winter 2013 | 67.4 | 54.7 | 122.0 | 55.2 | Li et al., 2016 |
| | | | | Spring 2014 | 78.6 | 51.0 | 129.6 | 60.6 | |
| | | | | Winter 2014 | 85.8 | 75.0 | 160.8 | 50.3 | |
| East-side of the Taiwan Strait | KH, TC, TP | Daily | 24 hrs | Winter 2013 | 48.0 | 22.98 | 71.0 | 67.6 | |
| | | | | Spring 2014 | 33.8 | 28.5 | 62.3 | 54.3 | |
| | | | | Winter 2014 | 53.4 | 54.6 | 108.0 | 49.5 | |

Islands represents the Xiaomen site at the Penghu Islands.



**Table 2. The measured water-soluble ionic species of PM$_{2.5}$ over sea and at the offshore islands during three cruise sampling campaigns.**

| Cruise Campaigns | Winter 2013 (W13) | | | | Spring 2014 (S14) | | | | | Winter 2014 (W14) | | | |
|---|---|---|---|---|---|---|---|---|---|---|---|---|---|
| Sampling Sites | Islands | Over Sea | | | Islands | Over Sea | | | | Islands | Over Sea | | |
| Species | W13 | W13C1 | W13C2 | W13C3 | S14 | S14C1 | S14C2 | S14C3 | S14C4 | W14 | W14C1 | W14C2 | W14C3 |
| PM$_{2.5}$ | 28.2 | 21.5 | 27.0 | 26.0 | 22.7 | 30.2 | 31.7 | 31.3 | 33.4 | 37.6 | 38.6 | 43.5 | 38.3 |
| SO$_4^{2-}$ | 4.8 | 3.6 | 4.1 | 4.2 | 4.3 | 4.9 | 4.4 | 4.6 | 5.0 | 6.5 | 5.2 | 5.6 | 5.3 |
| NO$_3^-$ | 3.1 | 2.5 | 2.6 | 2.9 | 2.3 | 3.0 | 2.9 | 2.9 | 3.2 | 4.0 | 3.2 | 3.4 | 3.4 |
| NH$_4^+$ | 1.2 | 0.8 | 1.0 | 1.2 | 1.1 | 2.0 | 1.9 | 1.9 | 2.1 | 2.2 | 2.4 | 2.6 | 2.3 |
| Cl$^-$ | 1.8 | 1.7 | 1.9 | 2.0 | 1.5 | 2.2 | 2.1 | 2.1 | 2.2 | 1.4 | 2.8 | 2.8 | 2.9 |
| Na$^+$ | 1.2 | 1.2 | 1.3 | 1.3 | 1.0 | 1.5 | 1.3 | 1.5 | 1.4 | 1.0 | 1.9 | 2.0 | 1.9 |
| K$^+$ | 1.5 | 0.5 | 0.7 | 0.5 | 1.1 | 0.4 | 0.5 | 0.3 | 0.5 | 0.9 | 0.5 | 0.7 | 0.6 |
| Mg$^{2+}$ | 0.5 | 0.6 | 0.6 | 0.6 | 0.5 | 0.7 | 0.7 | 0.6 | 0.6 | 0.6 | 0.8 | 0.8 | 0.6 |
| Ca$^{2+}$ | 0.3 | 0.3 | 0.4 | 0.3 | 0.5 | 0.4 | 0.4 | 0.4 | 0.5 | 0.6 | 0.5 | 0.6 | 0.6 |
| Sea salts[a] | 3.6 | 3.5 | 3.8 | 3.9 | 3.0 | 4.5 | 3.9 | 4.2 | 4.3 | 2.9 | 5.5 | 5.7 | 5.7 |
| Sea salts[b] | 3.0 | 2.9 | 3.2 | 3.3 | 2.5 | 3.7 | 3.3 | 3.5 | 3.6 | 2.4 | 4.6 | 4.8 | 4.8 |
| Sea salts[c] | 4.0 | 3.9 | 4.1 | 4.1 | 3.3 | 5.0 | 4.1 | 4.9 | 4.6 | 3.3 | 6.2 | 6.5 | 6.3 |
| Sea salts[a]/PM$_{2.5}$ | 12.7 | 16.1 | 13.9 | 15.0 | 13.2 | 14.7 | 12.4 | 13.4 | 12.8 | 7.7 | 14.3 | 13.1 | 14.8 |
| Sea salts[b]/PM$_{2.5}$ | 10.7 | 13.5 | 11.7 | 12.7 | 11.1 | 12.4 | 10.5 | 11.2 | 10.9 | 6.5 | 12.0 | 11.0 | 12.5 |
| Sea salts[c]/PM$_{2.5}$ | 14.0 | 18.1 | 15.2 | 15.9 | 14.7 | 16.5 | 12.9 | 15.5 | 13.8 | 8.7 | 16.1 | 14.9 | 16.3 |

Unit: μg m$^{-3}$; [a]: estimated by equation (1); [b]: estimated by equation (2) ; [c]: estimated by equation (3).

Islands represents the Xiaomen site at the Penghu Islands





**Table 3.** The concentrations of sea-salt and non-sea salt water-soluble ionic species of PM$_{2.5}$ over sea and at the offshore islands during three cruise sampling campaigns.

| Cruise Campaigns | Winter 2013 (W13) | | | | Spring 2014 (S14) | | | | | Winter 2014 (W14) | | | |
|---|---|---|---|---|---|---|---|---|---|---|---|---|---|
| Sampling Sites | Islands | Over Sea | | | Islands | Over Sea | | | | Islands | Over Sea | | |
| Species | W13 | W13C1 | W13C2 | W13C3 | S14 | S14C1 | S14C2 | S14C3 | S14C4 | W14 | W14C1 | W14C2 | W14C3 |
| PM$_{2.5}$ | 28.2 | 21.5 | 27.0 | 26.0 | 22.7 | 30.2 | 31.7 | 31.3 | 33.4 | 37.6 | 38.6 | 43.5 | 38.3 |
| nss-K$^+$ | 1.4 | 0.4 | 0.6 | 0.5 | 1.0 | 0.4 | 0.5 | 0.2 | 0.4 | 0.9 | 0.4 | 0.6 | 0.5 |
| nss-Mg$^{2+}$ | 0.4 | 0.4 | 0.4 | 0.5 | 0.4 | 0.5 | 0.5 | 0.5 | 0.5 | 0.5 | 0.6 | 0.6 | 0.4 |
| nss-Ca$^{2+}$ | 0.2 | 0.3 | 0.4 | 0.3 | 0.5 | 0.3 | 0.4 | 0.3 | 0.5 | 0.5 | 0.4 | 0.6 | 0.5 |
| nss-SO$_4^{2-}$ | 4.5 | 3.3 | 3.7 | 3.9 | 4.0 | 4.5 | 4.0 | 4.2 | 4.7 | 6.3 | 4.7 | 5.2 | 4.8 |
| nss-NO$_3^-$ | 2.7 | 2.0 | 2.2 | 2.6 | 2.0 | 2.5 | 2.7 | 2.3 | 2.8 | 3.6 | 2.5 | 2.6 | 2.7 |
| ss-K$^+$ | 0.1 | 0.0 | 0.1 | 0.1 | 0.0 | 0.1 | 0.1 | 0.1 | 0.1 | 0.0 | 0.1 | 0.1 | 0.1 |
| ss-SO$_4^{2-}$ | 0.3 | 0.3 | 0.3 | 0.3 | 0.3 | 0.4 | 0.3 | 0.4 | 0.4 | 0.3 | 0.5 | 0.5 | 0.5 |
| ss-Ca$^{2+}$ | 0.1 | 0.0 | 0.1 | 0.1 | 0.0 | 0.1 | 0.1 | 0.1 | 0.1 | 0.0 | 0.1 | 0.1 | 0.1 |
| ss-Mg$^{2+}$ | 0.1 | 0.1 | 0.2 | 0.2 | 0.1 | 0.2 | 0.2 | 0.2 | 0.2 | 0.1 | 0.2 | 0.2 | 0.2 |
| ss-NO$_3^-$ | 0.4 | 0.5 | 0.4 | 0.3 | 0.4 | 0.6 | 0.2 | 0.7 | 0.3 | 0.4 | 0.7 | 0.8 | 0.6 |
| meas-Cl$^-$ | 1.8 | 1.7 | 1.9 | 2.0 | 1.5 | 2.2 | 2.1 | 2.1 | 2.2 | 1.4 | 2.8 | 2.8 | 2.9 |
| meas-Na$^+$ | 1.2 | 1.2 | 1.3 | 1.3 | 1.0 | 1.5 | 1.3 | 1.5 | 1.4 | 1.0 | 1.9 | 2.0 | 1.9 |

Unit: µg m$^{-3}$; nss: non-sea salt ions; ss: sea salt ions; meas: measured ions
Islands represents the Xiaomen site at the Penghu Islands





**Table 4. Comparison of chloride deficit measured in marine aerosols at the coastal regions and in the remote oceans.**

| Regions/Seasons | Sample Size Range | Cl- Deficit Percentages | References |
|---|---|---|---|
| South China Sea/winter | Coarse and fine aerosols | 86% and 98% for fine-mode; 29% and 30% for coarse-mode | Hsu, et al., 2007 |
| East China Sea/(Korea) spring during Asian dust periods (ADS) and non-Asia dust periods (NADS) | Coarse and fine aerosols | 40% for fine-mode and 13% for coarse-mode during ADS periods, while during NADS periods 55% for fine-mode and 16% for coarse-mode | Park et al., 2004 |
| East coast of US/spring | Size-segregated aerosols | 14% | Keene and Savole, 1998 |
| Tropical northern Atlantic ocean/ spring | Coarse and fine aerosols | 29.7±9.9% for fine-mode and 11.9±13.3% for coarse-mode | Johansen et al., 2000 |
| Northern Indian Ocean/ late spring and summer | SW monsoon and inter-monsoon period | 3.5±6.3% in SW monsoon and 15.0±9.0 in the inter-monsoon periods | Johansen et al., 1999 |
| Tropical Arabian Sea/spring | Coarse and fine aerosols | 89±9% for fine-mode and 25.6±21.3% for coarse-mode. | Johansen and Hoffmann, 2004 |
| NW Mediterranean Sea | Size-segregated aerosols | 18.5±14.5% | Sellegri et al., 2001 |
| Southwestern coastal area of Taiwan Strait/ inland and offshore areas | Coarse and fine aerosols | 33.8±9.7% on inland and 22.5±6.9% on offshore for fine-mode; 33.8±9.1% on inland and 15.4±3.1% on offshore for coarse-mode. | Tsai et al., 2010 |
| West-side, east-side, offshore areas and sampling boat on sea of Taiwan Strait (2013-2015) | Fine aerosols | 40.1% on west-side, 41.2% on east-side, 27.8% on offshore area, and 23.5% on sea | This study |
| Offshore and sea samples at the Taiwan Strait/ winter and spring | Fine aerosols | 16.20% to 19.19% at offshore site, 7.56% to 20.41% over sea | This study |
| Coastal Antarctica | Size-segregated aerosols | 10–20% | Jourdain and Legrand, 2002; Rankin and Wolff, 2003 |



**Table 5. The percentages of metallic elements, Ni/Fe, and Ni/Al in PM$_{2.5}$ sampled in Asia.**

| Sampling Locations | Sampling Periods | Ti | Fe | Mn | Ca | Mg | K | Al | Pb | Ni | Cr | Zn | Ni/Fe | Ni/Al | References |
|---|---|---|---|---|---|---|---|---|---|---|---|---|---|---|---|
| Penghu Islands | Dec 2013-Dec 2014 | 1.2±0.6 | 1.3±0.4 | 0.7±0.3 | 2.0±0.9 | 2.3±1.2 | 2.2±1.6 | 3.0±1.2 | 0.7±0.1 | 0.9±0.3 | 0.6±0.1 | 0.6±0.2 | 0.7±0.3 | 0.3±0.1 | This study |
| Taiwan Strait | Dec 2013-Dec 2014 | 1.1±0.3 | 0.8±0.2 | 0.8±0.2 | 2.2±1.2 | 2.4±1.3 | 2.3±0.9 | 2.0±1.0 | 0.8±0.1 | 0.7±0.1 | 0.5±0.1 | 0.4±0.1 | 0.9±0.4 | 0.4±0.1 | This study |
| West-side of Taiwan Strait | June 2013-April-2015 | 1.1±0.4 | 0.9±0.3 | 0.6±0.1 | 2.0±1.0 | 1.9±0.9 | 2.5±1.3 | 1.2±0.5 | 0.6±0.3 | 0.6±0.2 | 0.4±0.2 | 0.7±0.4 | 0.7±0.2 | 0.5±0.2 | Li et al., 2016 |
| East-side of Taiwan Strait | June 2013-April-2015 | 1.0±0.3 | 1.0±0.5 | 0.7±0.2 | 2.2±1.3 | 2.4±1.3 | 3.1±1.1 | 1.1±0.6 | 0.7±0.3 | 0.7±0.2 | 0.6±0.1 | 0.7±0.3 | 0.7±0.3 | 0.6±0.2 | Li et al., 2016 |
| Over Sea of Taiwan Strait | June 2013-April-2015 | 1.2±0.2 | 1.0±0.5 | 0.8±0.2 | 2.8±1.7 | 3.1±1.8 | 4.5±2.0 | 1.2±0.3 | 1.0±0.3 | 0.8±0.2 | 0.9±0.3 | 0.8±0.4 | 0.8±0.3 | 0.7±0.2 | Li et al., 2016 |
| East China Sea | Spring 2011 | 0.8 | 5.3 | 0.0 | 6.5 | 2.9 | - | 12.4 | 0.0 | 0.0 | - | 0.2 | - | - | Zhao et al. 2015 |
| Taipei | Spring 2000 | 0.2 | 0.0 | 0.1 | 2.0 | 0.6 | - | 3.0 | 0.1 | - | - | 0.3 | - | - | Hsu et al., 2004 |
| Bohai Sea | 2006-2007 | 0.0 | 0.1 | 0.0 | 4.7 | 3.1 | - | 1.1 | 0.0 | - | - | 0.0 | - | - | Ji et al., 2011 |
| East China Sea | Spring 2008 | 0.1 | 0.8 | 0.0 | 1.4 | 0.8 | - | 0.8 | 0.0 | - | - | 0.9 | - | - | Wu et al., 2010 |





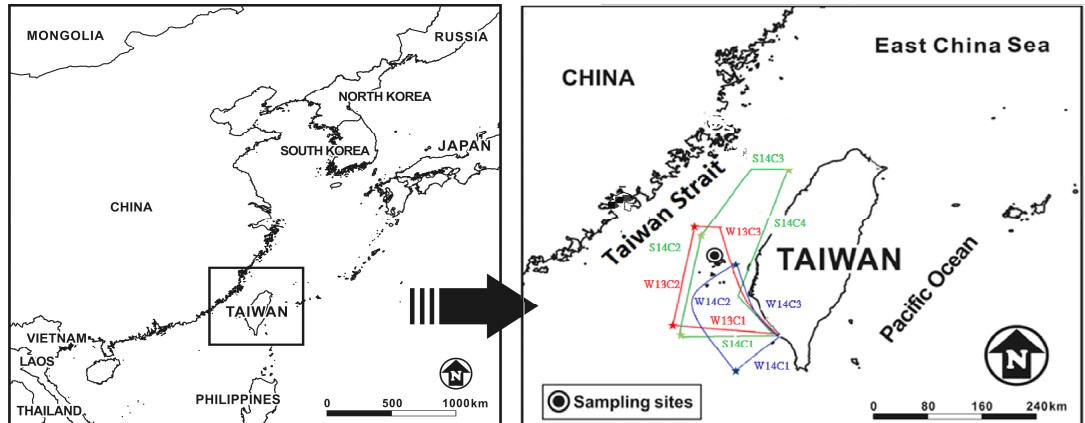

**Figure 1. The navigation routes and courses during three cruise sampling campaigns in the Taiwan Strait.**

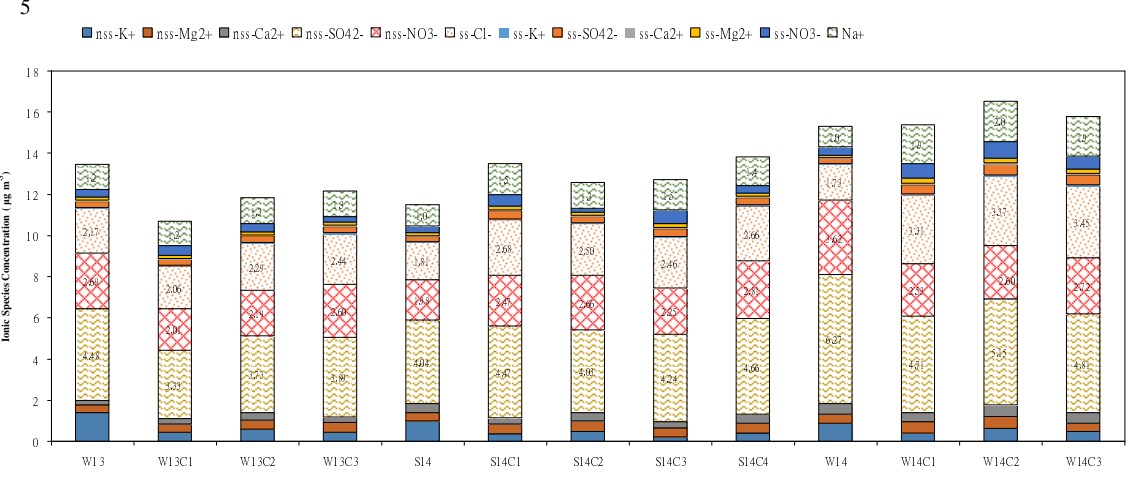

**Figure 2. The concentrations of sea salt- and non-sea salt-water-soluble ions and their**
20 **contribution to PM$_{2.5}$ sampled during the cruise sampling campaigns.**



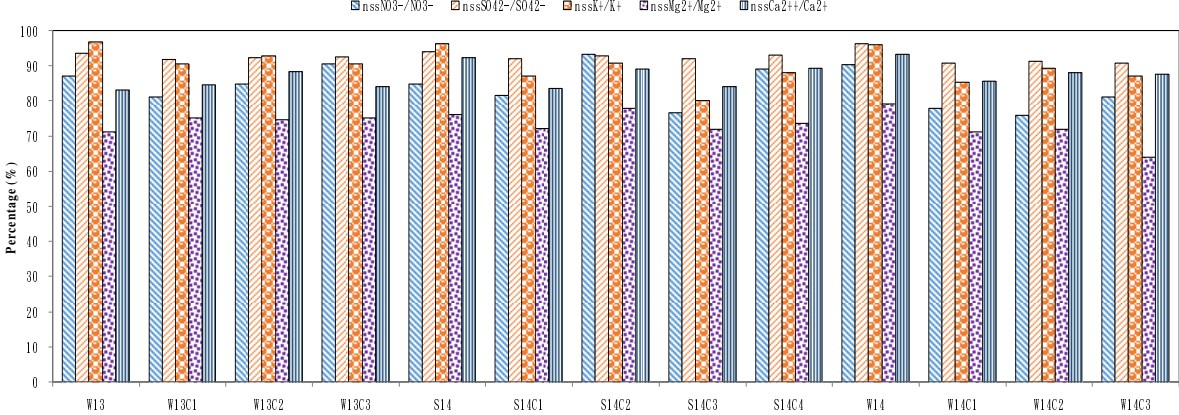

**Figure 3. The percentages of non-sea salt water-soluble ionic species in the measured water-soluble**
15 **ionic species of PM$_{2.5}$ sampled over sea and at the offshore islands during the cruise sampling**
**campaigns.**

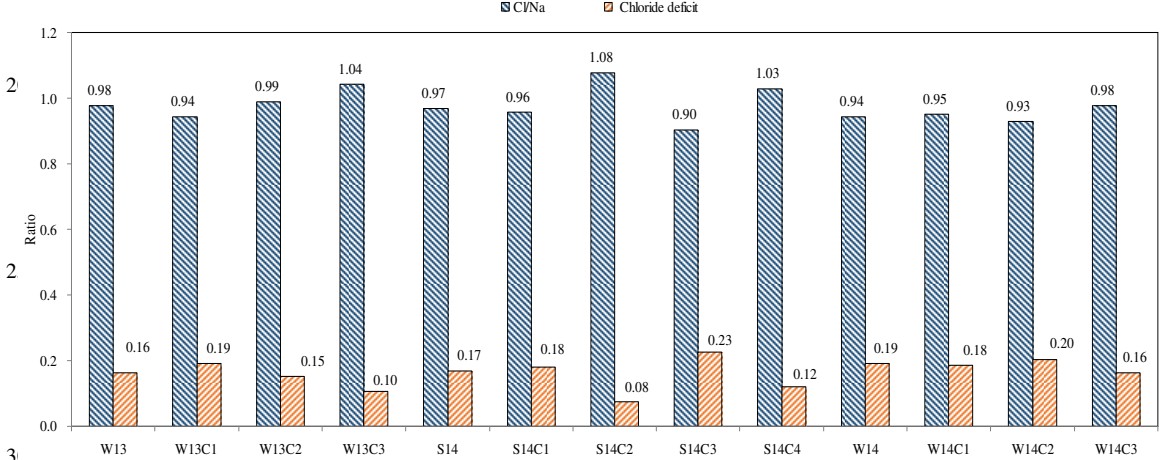

**Figure 4. The chloride deficit and the molar ratio of Cl$^-$/Na$^+$ in PM$_{2.5}$ over sea and at the offshore**
**islands during the cruise sampling campaigns.**





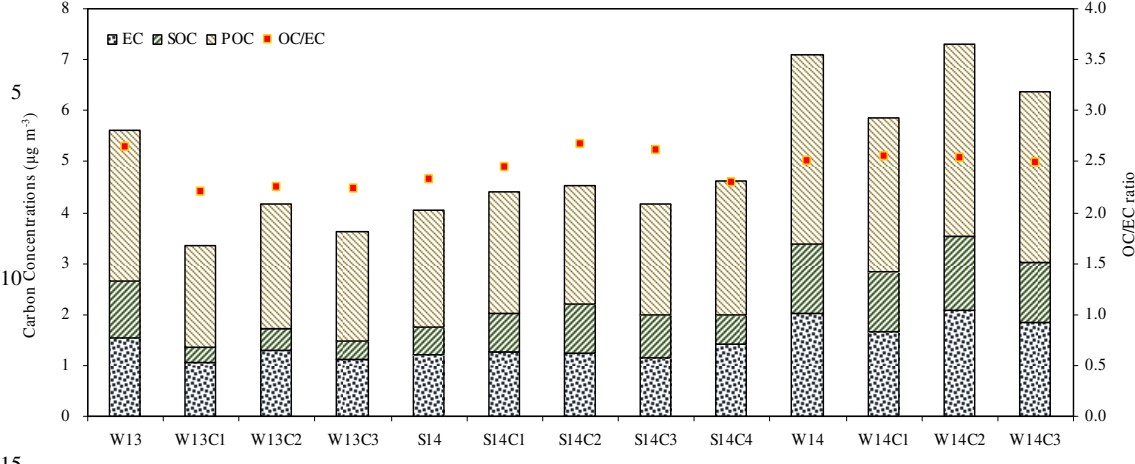

**Figure 5. The variation of estimated EC, POC, and SOC concentrations and OC/EC ratios in PM$_{2.5}$ over sea and at the offshore islands during the cruise sampling campaigns.**