# Peer review of "Chemical Characteristics of Marine Fine Aerosols over Sea and at Offshore Islands during Three Cruise Sampling Campaigns in the Taiwan Strait- Sea Salts and Anthropogenic Particles"

_Atmospheric Chemistry and Physics, 2016_

## Referee Comment (RC1) · Anonymous Referee #2 · 1 Sep 2016

Li et al.:
**Chemical Characteristics of Marine Fine Aerosols over Sea and at Offshore Islands during Three Cruise Sampling Campaigns in the Taiwan Strait- Sea Salts and Anthropogenic Particles**

**Review**

**General**
The paper presents concentrations of major ions, trace elements, organic and elemental carbon in PM2.5 saamples collected during cruises and on an island in the Taiwan Strait. The results are not particularily surprising but they might potentially be be publishable in ACP. However, in the present version of the paper there are serious problems and even some errors and it does not actually bring anything scientifically new information to the field. As it is now, it is mainly a data report and not a very good one even for such a purpose. I would recommend a major revision of the work and resubmission.

Uncertainty analyses are presented in section 2.3 but not detailed enough. The concentrations reported from filter samples involve various sources of uncertainty: uncertainties of flows, weighing, positive and negative artifacts, chemical analysis uncertainties, blank values, blank variability, detection limits, are different for different species. These should be discussed and given numerical values. A table within section 2.3 showing the concentrations (including gravimetric mass) of the analyzed species in the field blanks should be given.

Concentrations and mass balances are not properly presented in the paper. There should at least be a table or tables showing statistics of the concentrations of ions, trace elements and carbonaceous component and the percentages of the gravimetric mass. Now it is not possible to find out whether there was some mass missing compared to the gravimetric mass or whether the sum of species was higher than it. The Al concentration can be used for estimating the crustal mass from ~ 12 × Al(measured) since the ratio of Al to other elements in the Earth's crust is relatively constant, see for instance Wedepohl: Geochimica et cosmochimica Acta, 59(7), 1217-1232, 1995.

The use of "sea salts" in plural is a bit disturbing, in most cases it would be better to use the singular form "sea salt".

The ion data should be used for calculating something more than only chloride depletion. For instance ion balances by using the ion concentrations you would see whether the samples were acidic, alkaline or neutral. The ion data could be used for calculating enrichment factors enrichment factors compared to seawater composition and the trace element data for calculating enrichment factors compared to average crustal rock composition (for example Duce, et al. 1975, Science 10 : 59-61, Artaxo et al. 1992, Tellus B, 44: 318–334.; Wedepohl: Geochimica et cosmochimica Acta, 59(7), 1217-1232, 1995; Mishra et al, 2004, Atm.Env. 38, 4069-4084; etc. ).

Source area analysis would definitely be needed for this kind of a work. At least using air mass back trajectories or footprints with some meteorological model. Such are available and easy to use, for instance the HYSPLIT of NOAA can be used openly and it provides both back trajectories and dispersion modeling. They should be used to analyze what kind of concentrations or concentration

ratios or other derived properties – for instance ion balances by using all ions and ammonium to nss-sulfate ratios, or enrichment factors of trace elements, or OC-to-EC ratios or EC contributions – are associated with air masses from the different source areas.

**Detailed comments**

P3, L12. Write the manufacturer, type and model of the sampler. Also the filter type, filter manufacturer and other details, including diameter. The sampler was a high-volume one. What was the flow rate?

P3,L15: " **... we weighted the mass concentrations of both PM2.5 and PM2.5-10 after appropriate conditioning for each sample ...**". Describe the conditioning and weighing in more detail. At what humidity was the weighing done? Was RH measured? How long were the samples let stabilize at this RH before the actual weighing. The point is that quartz filters are notoriusly difficult for gravimetric analyses due to their hygroscopicity.

P3, L16-17: "**PM2.5-10 samples cannot be analyzed for chemical composition since they were contaminated with the oil coated on the surface of the impactors**". If the coarse-particle samples were contaminated, how can you be sure that the fine-particle filters remained clean?

P3, L26-28, "**The air quality sampling boat sailed to windward during the entire sampling voyage. Consequently, the winds were blown from the prow of the boat in order to avoid the intrusion of oil-burning particles emitted from the air quality sampling boat itself.**"
Hard to believe. I have also sampled on a ship and it is hard to avoid wind blowing from the wrong direction during such a long time. According to Figure 1 the ship sailed partially along direct tracks and turned sharply to the right at the locations marked by the stars. Did wind direction really turn so sharply? For example, I estimated the location of the ship during the cruise leg S14C1. According to the map in Fig.1 the ship started from south of Kaohsiung City and sailed westwards on 2014-04-10. I took a random point estimatedly on the cruise route by using Google Earth, wind data from NOAA-ARL web page and draw a wind rose (ready.arl.noaa.gov/READYamet.php) for the date 2014-04-10. According to that wind blew from the eastern sectors between NE and SE which means from behind the ship. On the other hand, if the ship was sailing at a high enough cruise speed, the relative wind direction may have been from the clean sector. For some other locations and times the model actually does show the ship was sailing against the wind. This kind of an analysis should be shown in time series plots of true and relative wind speed and direrction during the cruises. Was there any sector control?

P3, L29, "**Each sampling course was arranged to collect PM2.5 for continuous 8-12 hours...**". It is not quite clear whether there was only one sample taken during each course or were there more? For example during S14C4 was there only one or were there more samples taken? Explain this so that there is no ambiguity about it. And if there was more than one sample in each course then the values in the tables are averages or what?

P4, L8 " **Xiaomen site (23°38'47.1"N; 11930'31.6"E)**". There is obviously the degree sign (°) missing so that the coordinates should read (23°38'47.1"N; 119°30'31.6"E). But when I type in these coordinates in Google Earth the location seems to be not on any island but in the ocean to the southwest of the Xiaomen island and to the west of the northern part of Xi island. Check the coordinates and give them accurately.

Another small disturbing point is that there was only this one island measurement site but throughout the text it is written "at the islands". That is not justified, especially because the Xiaomen site was really close, only 60 m from the sea shore (sounds like a good location, by the way), so it is definitely less polluted by local sources and much more marine than some other locations on the Penghu islands. It is misleading to write that "on the islands" the concentrations were this or that. Change the text and tables all over so that you write Xiaomen or "on the island" instead of "the islands".

Further, a grammar issue related to the islands. Use the preposition "on" not "at". Check for instance http://ell.stackexchange.com/questions/8835/in-at-or-on-an-island

P4, L19-20: " **All quartz fiber filters were divided into four identical parts prior to the chemical analysis.**" What was the uncertainty associated with this division? Were each of these four pieces weighed also separately to find the accurate fraction of the filter that was used for each of the chemical analyses? Or was the division into four parts done only visually? Were the concentrations of the chemical species then scaled accordingly? If not, how do you know actually how big a part of the filter was used for each of the chemical analyses.

P4, L22 and L27. Some species were analyzed both by IC and by ICP-AES. How do these concentrations agree? For instance, at all the sites discussed in this paper Na is definitely only from the sea and the concentrations should be within the uncertainties the same. Ca, Mg, and K have also other sources. Discussion of the Na comparison would fit into the uncertainty section and the other comparisons to the trace element section. Make a figure with scatter plots of the concentrations of the species analyzed with these two methods.

P5,L29 – P6,L8. There is discussion that includes the coarse particles. Earlier, on P3, L16-17 it was written: "**$PM_{2.5-10}$ samples cannot be analyzed for chemical composition since they were contaminated with the oil coated on the surface of the impactors**". How is it then possible you discuss here also the $PM_{2.5-10}$?

P7,L31-33. There are 3 equations for calculating the concentration of sea salt. Only (1) makes sense. The major ions in sea salt are $Na^+$, $Cl^-$, $K^+$, $Mg^{2+}$, $Ca^{2+}$, $SO_4^{2-}$ and $HCO_3^-$, of which $Cl^-$ may get replaced. But the other major ions are there and it does not make any sense to calculate sea salt concentration by summing only sodium and chloride (Eq.(2)). Eq. (3) on the other hand does not make sense because in sea water nitrate is far from being a conservative compound, its concentrations vary a lot, and yet its contribution is very low. For instance Seinfeld and Pandis (2006) present in their Table 8: that the nitrate concentrations vary in a range of $3 \times 10^{-6}$ - $2 \times 10^{-3}$ % by weight. Measurements have shown that especially in the surface water nitrate concentrations are very low and one of the reasons is that nitrate is a nutrient used by marine biological organisms. Nitrate is a non-conservative tracer that is almost completely depleted in surface waters. So, it is very safe to claim that all nitrate in the filter samples analyzed in this work have come from other sources than sea water. So the only sensible equation for calculating seas salt concentration is (1). Consequently, the comparisons of the sea salt concentrations with the different equations are irrelevant and should be removed from the text, tables and figures.

P8, L6-7 " **Previous studies reported that the mass of sea salts can be adopted by the sum of $Cl^-$ and $Na^+$ (equation (2)) (Chow et al., 1996; Kim et al., 2000; Tsai et al., 2011; Han et al., 2003; Virkkula et al., 2006). .**.."
Let us check what these papers write about calculating sea salt.
-- Chow et al. (1996), p. 2106: "... sum of the soluble sodium and chloride to account for sea salt ..."

- Kim et al. (2000) don't tell at all how to calculate sea salt mass. On p. 2037 they write: "The $Cl^-$ to $Na^+$ ratio of sea water is 1.8; however, due to the loss of $Cl^-$ during transport, it is normally assumed to be 1.0." But nowhere in that paper they present how to calculate sea salt mass.
- Tsai et al.(2011) don't give any formula on how to calculate sea salt mass.
- Han et al. (2003) is a conference abstract not available in the open literature. I could not check it and it would be better not to refer to it at all.
- Virkkula et al. (2006) write on p.2: " Sea salt mass concentration was calculated from $Cl^-$ + $1.47Na^+$ where 1.47 is the seawater ratio of $(Na^+ + K^+ + Mg^{2+} + Ca^{2+} + SO_4^{2-} + HCO_3^-)/Na^+$ [Bates et al., 2001; Quinn et al., 2001]..." .
So, only Chow et al. (1996) write that sea salt mass could be calculated by summing up only sodium and chloride. But even if that is so in that paper it is definitely wrong, the other major sea-salt ions are present, as I wrote above.

P9,L2. Eq (8) is definitely wrong. As I wrote above, nitrate is not a seasalt compound. In the aerosol it is safe to claim that all nitrate is nss.

P9,L6 – 9. All sodium and chloride in aerosol are definitely sea salt on an island 60 m from the ocean shore and on the ship sailing on the ocean. So the texts ss-Cl and ss-Na should be removed. The concentrations of seasalt sulfate, seasalt magnesium, seasalt kalium and seasalt calcium are all calculated simply by multiplying observed sodium concentrations with the wellknown ratios of the ion X to sodium in seawater. The raw data for this calculation is only sodium concentration. So the ordering of ss ion concentrations in line 9 makes no sense at all. And there is an error even in that: ss Ca concentration should be higher than ss K.

P9,L29 "**Chloride deficit is a process**". No. Chloride deficit is a number calculated in Eq. (9). Chloride replacement is a process.

P10, L19-21 " **Previous study indicated that the aged nature of sea salt particles were about 150 km from the open sea, giving these particles enough time to react with atmospheric acidic gases (Virkkula et al., 2006).** " The referenced paper presented chloride depletion at a very clean Antarctic site so it is not comparable with the Taiwan Strait. The chloride replacement process can take place in a short period and distance if the concentrations of acidic gases are high.

P11, L11. " **… crustal elements (Al, Fe, Mg, K, and Ca), ...**" of these Mg, K, and Ca are also from sea salt. If you want to show the crustal elements only, do the seasalt correction.

P12, L17, Eq.(11) is strange. If you set in eq (10) it reduces to POC =EC×(OC/EC)pri so why don't you show it so? The method is very, very uncertain. The ratio (OC/EC)pri definitely varies according to burning material, burning temperature and other conditions. Then during transport organics condense on the particles. Your sampling sites are so far away from any sources that even the lowest OC/EC ratio at in the samples cannot represent the primary ratio at any conditions. Remove all text and results where you discuss SOC and POC. Just discuss OC, EC and particulate organic matter (POM). POM you would calculate by multipling OC with a factor that takes into account thpe amount of oxygen in organic aerosol. There are several references for this, look for them.

---

## Referee Comment (RC2) · Anonymous Referee #3 · 6 Oct 2016

Review for "Chemical Characteristics of Marine Fine Aerosols over Sea and at Offshore Islands during Three Cruise Sampling Campaigns in the Taiwan Strait- Sea Salts and Anthropogenic Particles" by Li et al. submitted to ACPD

This paper presents filter-based PM2.5 composition measurements area over Taiwan strait where is not well characterized and compares with the condition at an offshore island site in Penghu island. This is a high quality data set and an important contribution to the field. I do recognize that this is a data set over a large marine area with different cruises and as such it is difficult to analyze, but I do not think the analysis at present is

as thorough and coherent as it could be.

Overall comments: This paper presents almost all the compositions can be measured in PM2.5 in three courses over Taiwan strait, including the major ions, heavy metals and OCEC. The data quality is good however the overall impression is the authors just showed so much data there and did not find any impressed or interesting findings. I suggest to submit to other popular journals such as Atmospheric Environment or Atmospheric Research, unless the author can provide any interesting findings after further analysis.

Detailed comments: A The tables and figures A1: Units need to be added to the table 2 3 5 A2: Sample numbers should be added in table 1. Actually I was confused about the sampling method. Totally how many valid filters were collected in different cruises? If the filters of PM2.5-10 were contaminated, it can be moved to the supplemented materials as there are no important results. A3: Tables can contain large information than the one can be illustrated in the manuscript. The authors do not need to mention every data in the tables, but need to add some comparisons with the data in other literatures to rich the contents. Some interesting findings can be discovered during this process..

B Logic B1: The major discussion including seven sections, and less comparison with the data over other areas, which makes the reader feel that the author is just loading the data. It is better to find some internal connection between these data and name each section following the findings. Or the author could try 3.1.1 to including some sections into one section. e.g. 3.3, 3.4 and 3.5 are all about the sea salt particles, which can be in one section. B2: The outline really need to be reconstructed. I often have this problem with my paper that the closely related information is not discussed until much later in the paper. Please try to discuss, at least briefly, all the relevant information on a topic at one place. Otherwise, some issues sounds like mentioned several times. Language or content can be more condensed.

C Detailed comments C1: The weather condition should be also mentioned at first as the sampler number is limited and the reader need some general idea on the background air mass condition. C2: Page 6 line 30 section 3.2: It should be ammonium-poor area that (NH4)2SO4 is not favored. NH4HSO4 is more likely. C3: Page 8 line 9. the data obtained by equation(1) is in the middle level of these three results, however it is not a reason that "the most accurate method to estimate the sea salt concentrations was equation (1)". C4: Page 9 line 10-15 about the anthropogenic particle influence, it can be one important topic in this manuscript. Suggest the authors make two major concern: anthropogenic source and sea salts Cl deficit. Besides Section 3.5 title is missing in the manuscript. C5:Page 12 line 30, an accurate (OC/EC)pri value used in this study should be mentioned in the manuscript. The discussion of OCEC is really poor.

---

## Author Comment (AC1) · 16 Nov 2016

Referee comment 1

(1)Uncertainty analyses are presented in section 2.3 but not detailed enough. The concentrations reported from filter samples involve various sources of uncertainty: uncertainties of flows, weighing, positive and negative artifacts, chemical analysis uncertainties, blank values, blank variability, detection limits, are different for different species. These should be discussed and given numerical values. A table within Section 2.3 showing the concentrations (including gravimetric mass) of the analyzed species in the

field blanks should be given.

» Thanks for the comment. The results of QA/QC have been summarized in Table 1 per request. (see P25).

(2) Concentrations and mass balances are not properly presented in the paper. There should at least be a table or tables showing statistics of the concentrations of ions, trace elements and carbonaceous component and the percentages of the gravimetric mass. Now it is not possible to find out whether there was some mass missing compared to the gravimetric mass or whether the sum of species was higher than it. The Al concentration can be used for estimating the crustal mass from $\sim 12 \times$ Al(measured) since the ratio of Al to other elements in the Earth's crust is relatively constant, see for instance Wedepohl: Geochimica et cosmochimica Acta, 59(7), 1217-1232, 1995.

» Thanks for the comment. We have revised Table 3 per request (see Table 3, P27). Moreover, we have also added a Section of "3.5 Reconstruction of PM2.5 over Sea and at the Offshore Islands," per request. PM2.5 was estimated by material balance equation for gravimetric mass (Chow et al., 1996). In this study, PM2.5 was summed by nine major chemical compositions: nitrate ($NO_3-$), sulfate ($SO_4{}^{2-}$), ammonium ($NH_4+$), chloride ($Cl-$), organic matter (OM), elemental carbon (EC), crustal materials (CM), sea salt, and others. Organic material (OM) was estimated from an organic carbon (OC) multiplier (f) that accounts for unmeasured hydrogen (H), oxygen (O), nitrogen (N), and sulfur (S) in organic compounds (Chow et al., 1996). Multipliers of 1.4 to 1.8 have been found to best represent the complex mixture of organic molecules in OM (POM) (Chow et al., 1996). A factor of 1.6 for converting OC to OM was used in this study. Crustal materials can be estimated using a method reported by Wedepohl (1995) (Crustal materials= 12*[Al]), while sea salt was estimated by equation (3). Therefore, PM2.5 concentrations were reconstructed by the following equation,

Table 8 compares the major chemical components of PM2.5 at the coastal sites around the Taiwan Strait and East China Sea. Consistent with previous studies, organic materials (OM), SO42-, NO3-, and crustal materials (CM) were the major components for the reconstruction of PM2.5 concentration. The results indicated that SO42-, NO3-, crustal materials, and organic materials are important components in PM2.5 at all coastal sites. The contribution of SO42-, NO3-, and organic materials were similar to other coastal sites. (see P15, L17- L31).

(3) The use of "sea salts" in plural is a bit disturbing, in most cases it would be better to use the singular form "sea salt".

» Thanks for the comment. We have replaced "sea salts" with "sea salt" throughout the entire manuscript per request.

(4) The ion data should be used for calculating something more than only chloride depletion. For instance ion balances by using the ion concentrations you would see whether the samples were acidic, alkaline or neutral. The ion data could be used for calculating enrichment factors enrichment factors compared to seawater composition and the trace element data for calculating enrichment factors compared to average crustal rock composition (for example Duce, et al. 1975, Science 10 : 59-61, Artaxo et al. 1992, Tellus B, 44: 318–334.; Wedepohl: Geochimica et cosmochimica Acta, 59(7), 1217-1232, 1995; Mishra et al, 2004, Atm.Env. 38, 4069-4084; etc. ).

» Thanks for the comments. The ionic balance (i.e. A/C) of ionic species ranged from 0.7 to1.0, including that PM2.5 were acidic particles. (see Table 3, P27 and P9, L28-29). In previous studies, the particle-induced X-ray emission or proton-induced X-ray emission (PIXE) was used to determine the elemental characteristics of atmospheric particles (Artaxo et al. 1992). In this study, we analyzed ionic species and metallic content by IC and ICP-AES. Comparing different methods for analyzing chemical composition is difficult to calculate the enrichment factors of ionic species (e.g. Cl-), respectively. In this study, same methods were applied to analyze PM2.5 samples collected at the target region, which should be comparable for the enrichment factors of PM2.5 over sea and at the offshore islands. Thus, we used the presence of certain

metallic elements in aerosols surrounding the Taiwan Strait primarily due to natural or anthropogenic processes in the target regions. (see P16, L1-L10). Additionally, we have added a Section of "3.6 Enrichment Factors of PM2.5 over Sea and at the Offshore Islands," per request. The EF values of metallic elements in PM2.5 over sea and at the offshore islands are depicted in Figure 6. The order of the EF values for various metallic elements had quite similar trend no matter where atmospheric PM2.5 were sampled. For least ten measured metallic elements, their EF values were in the range of 0.1 to 10000 and highly relevant. Trace elements Ni and Cr were highly enriched (100<EF<10000) in PM2.5, while Mn and Pb were moderately enriched (10<EF<100) at all sites around the Taiwan Strait. Previous studies reported that metallic elements with EF>10 have an important proportion of non-crustal sources and that a variety of emission sources could contribute to their loading in the ambient air. The EF values of crustal elements Mg, K, Ca, and Fe in PM2.5 ranged from 1 to 10 over sea and at the offshore islands, and their EF values were quite consistent for fine particles sampled at different sites. It suggested that these crustal elements were likely originated from same natural sources and had no enrichment in PM2.5. In comparison, high EF values of Ni, Cr, Mn, and Pb in the range of 10-10000 suggested that these trace elements were mainly originated from anthropogenic sources. Previous study reported that metallic elements Cr and Ni in PM2.5 were mainly from anthropogenic combustion sources, while Cr and Ni in PM2.5-10 had more soil-related origins (Chow et al., 1995). (see P16, L16-L27).

(5) Source area analysis would definitely be needed for this kind of a work. At least using air mass back trajectories or footprints with some meteorological model. Such are available and easy to use, for instance the HYSPLIT of NOAA can be used openly and it provides both back trajectories and dispersion modeling.

» Thanks for the comments. In order to identify the predominant sources of air pollutants, backward trajectory has been widely used to trace the transport routes of air masses (Li et al., 2016) (see P7, L4-L6). Previous study reported that the level of atmospheric PM2.5 is affected by meteorological condition, thus PM2.5 concentrations in spring and winter was much higher than those in fall and summer in the Taiwan Strait (Li et al., 2016b). However, the results of backward trajectory analysis were not shown in this manuscript. Our previous study indicated that the corresponding trajectories were clustered into three major transport routes according to their airflow directions and regions through which air masses traveled toward the Taiwan Strait (Li et al., 2016b). During the consecutive courses in the winter of 2013 (W3), air masses originated from Mongolia were transported across the northern, central, and southeastern China. During the consecutive courses in the spring of 2014 (S14), air masses originated from northern and northeastern China were transported through the coastal regions of central China, East China Sea, and southeastern China toward the Taiwan Strait. During the consecutive courses in the winter of 2014 (W14), air masses originated from northern and northeastern China are transported through the coastal regions of central and southeastern China toward the Taiwan Strait. Results from backward trajectories showed that the concentrations of PM2.5 blown from the north were generally higher than those from the south. PM2.5 samples were collected over sea and at the offshore site during the high pollution seasons in this study. In this study, 72-hour backward trajectories ending at the Penghu Islands at the altitudes of 100, 350, and 500 m above sea level, respectively, were simulated to represent air masses toward the Taiwan Strait. Air masses originated from Mongolia were transported through northern and central China, and finally across the East China Sea to the offshore site during the three sampling cruises. The results indicated that anthropogenic chemical species were evenly dispersed over sea for the same trajectory during the air pollution episodes, causing the carbonaceous species stably distributed over sea and at the offshore site. (see Section 2.4, P7, L11-15, and Section 3.2, P8, L17-P9, L8).

(6) They should be used to analyze what kind of concentrations or concentration ratios or other derived properties – for instance ion balances by using all ions and ammonium to nss-sulfate ratios, or enrichment factors of trace elements, or OC-to-EC ratios or EC contributions – are associated with air masses from the different source areas.

» Thanks for the comments. We have added a Section of "3.4 Distribution and Source Indicators of PM2.5 Chemical Composition over Sea and at the Offshore Islands," to describe the distribution of the mass percentage of chemical composition of PM2.5 over sea and on the offshore islands. The results of the distribution percentage of chemical characteristics to PM2.5 during the three courses over Taiwan Strait indicated that water-soluble ionic species, metallic elements, and carbonaceous content accounted for 46.1-52.0%, 14.7-19.3%, and 14.0-19.9% of PM2.5 during the sampling courses in the winter of 2013, respectively; 44.4-54.1%, 13.1-15.7%, and 13.3-17.9% during the sampling courses in the spring of 2014, respectively; 42.7-45.8%, 12.3-13.4%, and 15.2-18.8% during the sampling courses in the winter of 2014, respectively. The results indicated that the distribution of water-soluble ionic species and carbonaceous contents on the offshore islands were generally higher than those over sea. Previous study reported that the emissions of huge amounts of particulates from various sources (e.g., textile plants at the Jinjing River Basin) could result in the higher percentages of ionic and carbonaceous contents at the Taiwan Strait (Li et al., 2016a). The results were close to those at the Penghu site located at an offshore island. (see P14, L18-L28) There are several ratios of chemical species can be used as valuable indicators to appoint atmospheric particles from specific sources (Cao et al., 2012; Arimoto et al., 1992). Previous researches reported that the mass ratios of EC/TC, K+/TC, and TC/SO42- can be used to identify the sources from biomass burning (VanCuren, 2013). When the mass ratio of EC/TC ranges from 0.1 to 0.2, K+/TC ranges from 0.5 to 1.0, and TC/SO42- ranges from 6 to 15, it suggests that the sources were mainly contributed from biomass burning (VanCuren, 2013). The mass ratios of NO3-/nss-SO42- have also been used to evaluate the contributions from stationary and mobile sources (Arimoto et al., 1992). The mass ratios of NO3-/nss-SO42- higher than unity indicated that the sources of particles were mainly from mobile sources. Conversely, the mass ratios of NO3-/nss-SO42- lower than unity suggested that the sources of particles came mainly from stationary sources. Table 7 compares the mass ratios of major chemical species over sea and at the offshore islands. The mass ratios of

EC/TC, K+/TC, NO3-/nss-SO42-, and TC/SO42- during the three courses over Taiwan Strait ranged from 0.28±0.01 to 0.30±0.02 in the winter of 2013 (W13), 0.10±.02 to 0.17±.05, 0.66±0.05 to 0.71±0.03 in the spring of 2014 (S14), and 0.94±0.05 to 1.18±0.08 in the winter of 2014 (W14), respectively, while similar trends reported at the southeastern coastline of the Taiwan Strait (Li et al., 2016a). Previous studies reported that high SO42- and NO3- concentrations observed at the Penghu site and the Kaohsiung site were mainly from stationary sources due to burgeoning industrial development in the southwestern coastal region of Taiwan (Tsai et al., 2011; Li et al., 2016a). According to the reports from VanCuren (2003), atmospheric aerosols with high nss-SO42-/NO3- ratios were attributed mainly from stationary sources. (see P14, L21-P15, L16). Moreover, the mass concentrations of PM2.5 and their water-soluble ionic species, metallic elements, and carbonaceous contents for clustered air mass trajectories toward the Taiwan Strait have been discussed and reported in our previous literature (Li et al., 2016b).

Detailed Comments (7) P3, L12. Write the manufacturer, type and model of the sampler. Also the filter type, filter manufacturer and other details, including diameter. The sampler was a high-volume one. What was the flow rate?

» Thanks for the comment. The high-volume air sampler (TE-6001) was used to collect PM2.5 with a sampling flow rate of 1.47 m3/min passing through a PM2.5 selective inlet. As the particulates travel through the PM10 size selective inlet the larger particulates are trapped inside of the inlet as the smaller PM2.5 particulates continue to travel through the inlet and are collected on the 8" x 10" quartz fiber filter manufactured by Pall Corporation. This method was complied with the sampling method of NIEA A102.12A similar to USEPA Method IO-2.1. Quartz fiber filter was selected for this study because we conducted the chemical analysis of water-soluble ions, metallic elements, and carbonaceous content. Before weighing, the quartz fiber filters were conditioned in a desiccator at temperatures of 20-25°C and relative humidity (RH) of 35-45% for 48 hours (see Section 2.3, P6, L6- L13).

(8) P3, L15: ".... we weighted the mass concentrations of both PM2.5 and PM2.5-10 after appropriate conditioning for each sample ...". Describe the conditioning and weighing in more detail. At what humidity was the weighing done? Was RH measured? How long were the samples let stabilize at this RH before the actual weighing. The point is that quartz filters are notoriously difficult for gravimetric analyses due to their hygroscopicity.

» Thanks for the comments. Quartz fiber filter was selected for this study because we conducted the chemical analysis of water-soluble ions, metallic elements, and carbonaceous content. Before weighing, the quartz fiber filters were conditioned in a desiccator at temperatures of 20-25°C and relative humidity (RH) of 35-45% for 48 hours. After conditioning, the filters were then weighed by a microbalance (Sartorius MC 5) with the precision of 1 $\mu$g to determine the mass of PM2.5. The filters were stored in a weighing chamber at temperatures of 20-25°C and relative humidity of 35-45% (see Section 2.3, P6, L10- L15).

(9) P3, L16-17: "PM2.5-10 samples cannot be analyzed for chemical composition since they were contaminated with the oil coated on the surface of the impactors". If the coarse-particle samples were contaminated, how can you be sure that the fine-particle filters remained clean?

» Thanks for the comments. We have revised the sentence as "In the sampler, an adapter is placed into the model TE-6001 sampler in lieu of an existing PM10 fractionator. The adapter has a plate that contains multiple impactors for collecting particles larger than PM2.5 on a slotted quartz fiber filter. PM2.5 is then passed through the impactor and collected on a quartz fiber filter. After sampling, the concentrations of PM2.5 and PM2.5-10 were determined by weighting the quartz fiber filter and the slotted quartz fiber filter, respectively. Due to the difficulty of identically dividing the slotted quartz fiber filter, we thus only analyzed the chemical compositions of PM2.5 (see Section 2.3, P6, L19-L24).

[Figure]

(10) P3, L26-28, "The air quality sampling boat sailed to windward during the entire sampling voyage. Consequently, the winds were blown from the prow of the boat in order to avoid the intrusion of oil-burning particles emitted from the air quality sampling boat itself." Hard to believe. I have also sampled on a ship and it is hard to avoid wind blowing from the wrong direction during such a long time. According to Figure 1 the ship sailed partially along direct tracks and turned sharply to the right at the locations marked by the stars. Did wind direction really turn so sharply? For example, I estimated the location of the ship during the cruise leg S14C1. According to the map in Fig.1 the ship started from south of Kaohsiung City and sailed westwards on 2014-04-10. I took a random point estimatedly on the cruise route by using Google Earth, wind data from NOAAARL web page and draw a wind rose (ready.arl.noaa.gov/READY amet.php) for the date 2014-04-10. According to that wind blew from the eastern sectors between NE and SE which means from behind the ship. On the other hand, if the ship was sailing at a high enough cruise speed, the relative wind direction may have been from the clean sector. For some other locations and times the model actually does show the ship was sailing against the wind. This kind of an analysis should be shown in time series plots of true and relative wind speed and direction during the cruises. Was there any sector control?

» Thanks for the comments. In this study, the air quality sampling boat sailed continuously windward in a speed of 10 knots per hour during the sampling cruise periods. Atmospheric fine particles were sampled at the fore of the uppermost deck on the shipboard with a high-volume sampler during the voyage to prevent the interferences from the exhaust gases of the air quality sampling boat itself as the chimney of the sampling boat is located at the poop deck. After carefully checking with the records of wind speeds and wind direction in the sampling boat, it showed that the prevailing wind came mainly from the northeast ($25°$-$53°$), in 2014-04-10, which did not blow the plume emitted from the chimney of the sampling boat to the PM2.5 sampler during the cruise sampling periods. (see P4, L3-L10).

(11) P3, L29, "Each sampling course was arranged to collect PM2.5 for continuous 8-12 hours...". It is not quite clear whether there was only one sample taken during each course or were there more? For example during S14C4 was there only one or were there more samples taken? Explain this so that there is no ambiguity about it. And if there was more than one sample in each course then the values in the tables are averages or what?

» Thanks for the comments. In this study, only one fine particle sample was collected during each sampling course. (see P4, L11-L12 and Table 3, P26).

(12) P4, L8 " Xiaomen site (23°38'47.1"N; 11930'31.6"E)". There is obviously the degree sign (°) missing so that the coordinates should read (23°38'47.1"N; 119°30'31.6"E). But when I type in these coordinates in Google Earth the location seems to be not on any island but in the ocean to the southwest of the Xiaomen island and to the west of the northern part of Xi island. Check the coordinates and give them accurately. Another small disturbing point is that there was only this one island measurement site but throughout the text it is written "at the islands". That is not justified, especially because the Xiaomen site was really close, only 60 m from the sea shore (sounds like a good location, by the way), so it is definitely less polluted by local sources and much more marine than some other locations on the Penghu islands. It is misleading to write that "on the islands" the concentrations were this or that. Change the text and tables all over so that you write Xiaomen or "on the island" instead of "the islands".

» Thanks for the comments. We have corrected the coordinates of Xiaomen site (23°33'41"N; 119°35'10"E) and replaced "the islands" with "on the island" throughout the entire manuscript per request. (see P4, L22, and Table 2, P26)

Further, a grammar related to the islands. Use the preposition "on" not "at". Check for instance http://ell.stackexchange.com/questions/8835/in-at-or-on-an-island

» Thanks for the comments. We have revised the preposition "at" with "on" the islands

throughout the entire manuscript per request.

(13) P4, L19-20: "All quartz fiber filters were divided into four identical parts prior to the chemical analysis." What was the uncertainty associated with this division? Were each of these four pieces weighed also separately to find the accurate fraction of the filter that was used for each of the chemical analyses? Or was the division into four parts done only visually? Were the concentrations of the chemical species then scaled accordingly? If not, how do you know actually how big a part of the filter was used for each of the chemical analyses.

» Thanks for the comments. In this study, we divided each filter into four identical parts for analyzing the chemical composition of PM2.5. After dividing, each of the identical parts was weighed separately to make sure the accurate fraction of the filter that was used for further chemical analyses. After analyzing the chemical composition, the concentrations of the chemical species were then quantified accordingly. (see P.5, L2-L5).

(14) P4, L22 and L27. Some species were analyzed both by IC and by ICP-AES. How do these concentrations agree? For instance, at all the sites discussed in this paper Na is definitely only from the sea and the concentrations should be within the uncertainties the same. Ca, Mg, and K have also other sources. Discussion of the Na comparison would fit into the uncertainty section and the other comparisons to the trace element section. Make a figure with scatter plots of the concentrations of the species analyzed with these two methods.

» Thanks for the comments. The water-soluble ionic species of PM2.5 were analyzed by IC, while the metallic contents were analyzed by ICP in this study. This study aims to characterize the chemical composition (especially water-soluble ionic species) of marine fine particles (PM2.5) in the atmosphere during the highly polluted seasons and to investigate how importance of sea salt and anthropogenic particles influenced PM2.5 in the Taiwan Strait. Therefore, the discussion of chemical species in PM2.5 focused on

water-soluble ionic species in the manuscript. Furthermore, the comparison of metallic content in PM2.5 focused on anthropogenic sources from their original region in the Taiwan Strait. A scatter diagram has been plotted in Figure 2 shown below per request. It showed that the concentrations of metallic contents (Mg, K, and Ca) analyzed by ICP-AES were always higher than those of water-soluble ionic species analyzed by IC (see P5, L15-L22).

(15) P5, L29–P6, L8. There is discussion that includes the coarse particles. Earlier, on P3, L16-17 it was written: "PM2.5-10 samples cannot be analyzed for chemical composition since they were contaminated with the oil coated on the surface of the impactors". How is it then possible you discuss here also the PM2.5-10?

» Thanks for the comments. We have revised the sentence as "In the sampler, an adapter is placed in the model TE-6001 sampler in lieu of the existing PM10 fraction. The adapter has a plate that contains multiple impactors for collecting particles larger than PM2.5 on a slotted quartz fiber filter. PM2.5 is then passed through the impactor and collected on a hi-volume quartz fiber filter. After sampling, the concentrations of PM2.5 and PM2.5-10 were then determined by weighing the quartz fiber filter and the slotted quartz fiber filter, respectively. Due to the difficulty of identically dividing the slotted quartz fiber filter, we thus solely analyzed the chemical compositions of PM2.5. (see P6, L19-L24)

(16) P7, L31-33. There are 3 equations for calculating the concentration of sea salt. Only (1) makes sense. The major ions in sea salt are Na+, Cl-, K+, Mg2+, Ca2+, SO42- and HCO3-, of which Cl- may get replaced. But the other major ions are there and it does not make any sense to calculate sea salt concentration by summing only sodium and chloride (Eq.(2)). Eq. (3) on the other hand does not make sense because in sea water nitrate is far from being a conservative compound, its concentrations vary a lot, and yet its contribution is very low. For instance Seinfeld and Pandis (2006) present in their Table 8: that the nitrate concentrations vary in a range of $3 \times 10\text{-}6$ - $2 \times 10\text{-}3$ % by weight. Measurements have shown that especially in the surface water

nitrate concentrations are very low and one of the reasons is that nitrate is a nutrient used by marine biological organisms. Nitrate is a non-conservative tracer that is almost completely depleted in surface waters. So, it is very safe to claim that all nitrate in the filter samples analyzed in this work have come from other sources than sea water. So the only sensible equation for calculating seas salt concentration is (1). Consequently, the comparisons of the sea salt concentrations with different equations were irrelevant and should be removed from the text, tables, and figures.

» Thanks for the comments. We have removed the comparisons of the sea salt concentrations estimated by equations (2) and (3) per request. Additionally, we rearranged the Sections in the manuscript and further revised the Tables and Figures as well.

(17) P8, L6-7 " Previous studies reported that the mass of sea salts can be adopted by the sum of Cl- and Na+ (equation (2)) (Chow et al., 1996; Kim et al., 2000; Tsai et al., 2011; Han et al., 2003; Virkkula et al., 2006). ..." Let us check what these papers write about calculating sea salt. Chow et al. (1996), p. 2106: "... sum of the soluble sodium and chloride to account for sea salt ..." - Kim et al. (2000) don't tell at all how to calculate sea salt mass. On p. 2037 they write: "The Cl– to Na+ ratio of sea water is 1.8; however, due to the loss of Cl– during transport, it is normally assumed to be 1.0." But nowhere in that paper they present how to calculate sea salt mass. - Tsai et al.(2011) don't give any formula on how to calculate sea salt mass. - Han et al. (2003) is a conference abstract not available in the open literature. I could not check it and it would be better not to refer to it at all. - Virkkula et al. (2006) write on p.2: " Sea salt mass concentration was calculated from Cl- +1.47Na+ where 1.47 is the seawater ratio of (Na+ + K+ + Mg2+ + Ca2+ + SO42- + HCO3-)/Na+ [Bates et al., 2001; Quinn et al., 2001]..." . So, only Chow et al. (1996) write that sea salt mass could be calculated by summing up only sodium and chloride. But even if that is so in that paper it is definitely wrong, the other major sea-salt ions are present, as I wrote above.

» Thanks for the comment. We have revised the method to estimate the concentrations of sea salt in the manuscript per request (see Section 3.3.1, P10, L25-L27).

(18) P9, L2. Eq (8) is definitely wrong. As I wrote above, nitrate is not a sea salt compound. In the aerosol it is safe to claim that all nitrate is nss.

» Thanks for the comment. We have removed the original Equation (8) and have revised Figures 3 and 4, and Table 4 in the manuscript per request (see Section 3.3.2, P11, L16-L19).

(19) P9, L6-9. All sodium and chloride in aerosol are definitely sea salt on an island 60 m from the ocean shore and on the ship sailing on the ocean. So the texts ss-Cl and ss-Na should be removed. The concentrations of sea salt sulfate, sea salt magnesium, sea salt kalium and sea salt calcium are all calculated simply by multiplying observed sodium concentrations with the well known ratios of the ion X to sodium in seawater. The raw data for this calculation is only sodium concentration. So the ordering of ss ion concentrations in line 9 makes no sense at all. And there is an error even in that: ss Ca concentration should be higher than ss K.

» Thanks for the comment. We have removed the sentence regarding the order of sea salt ion concentrations per request. (see Section 3.3.2, P11, L22-L25).

(20) P9,L29 "Chloride deficit is a process". No. Chloride deficit is a number calculated in Eq. (9). Chloride replacement is a process. » Thanks for the comment. We have revised the sentence as "Chloride replacement is the process by……" per request. (see P17, L2)

(21) P10, L19-21 " Previous study indicated that the aged nature of sea salt particles were about 150 km from the open sea, giving these particles enough time to react with atmospheric acidic gases (Virkkula et al., 2006). " The referenced paper presented chloride depletion at a very clean Antarctic site so it is not comparable with the Taiwan Strait. The chloride replacement process can take place in a short period and distance if the concentrations of acidic gases are high.

» Thanks for the comment. We have deleted the referred sentence (Virkkula et al.,

2006) from the manuscript per request.

(22) P11, L11. " ... crustal elements (Al, Fe, Mg, K, and Ca), ..." of these Mg, K, and Ca are also from sea salt. If you want to show the crustal elements only, do the sea salt correction.

» Thanks for the comments. Table 5 summarizes the concentrations of metallic elements in PM2.5. It showed that crustal and sea salt elements (Al, Fe, Mg, K and Ca) were higher than those of anthropogenic elements (Zn, Ni, and Pb). Accordingly, the sentence has been revised as, "Table 5 shows that the most abundant metallic elements of PM2.5 were crustal and sea salt elements (Na, Al, Fe, Mg, K, and Ca) and followed by anthropogenic elements (Zn, Ni, and Pb)" (see P12, L19-L20).

(23) P12, L17, Eq.(11) is strange. If you set in eq (10) it reduces to POC =EC×(OC/EC)pri so why don't you show it so? The method is very, very uncertain. The ratio (OC/EC)pri definitely varies according to burning material, burning temperature and other conditions. Then during transport organics condense on the particles. Your sampling sites are so far away from any sources that even the lowest OC/EC ratio at in the samples cannot represent the primary ratio at any conditions. Remove all text and results where you discuss SOC and POC. Just discuss OC, EC and particulate organic matter (POM). POM you would calculate by multiplying OC with a factor that takes into account the amount of oxygen in organic aerosol. There are several references for this, look for them.

» Thanks for the comments. We have revised the Section 3.3.4 "Carbonaceous Contents of PM2.5 over Sea and at the Offshore Islands," per request. (see P13, L15-P14, L20). Moreover, in this study, particulate organic matter (POM) was estimated by 1.6*[OC] as shown in Table 8.

Referee comment 2

Detailed comments: A. The tables and figures A1: Units need to be added to the table

2 3 5.

» Thanks for the comments. We have added the units in Tables 3, 4, and 5 per request. (see P26-P28)

A2: Sample numbers should be added in Table 1. Actually I was confused about the sampling method. Totally how many valid filters were collected in different cruises? If the filters of PM2.5-10 were contaminated, it can be moved to the supplemented materials as there are no important results.

» Thanks for the comments. In this study, only one fine particle sample was collected during each sampling course. The number of PM2.5 samples over sea and at the offshore islands during three cruises sampling campaigns was 4, 5, and 4, respectively (see P4, L11-L12 and Table 3, L27).

A3: Tables can contain large information than the one can be illustrated in the manuscript. The authors do not need to mention every data in the tables, but need to add some comparisons with the data in other literatures to rich the contents. Some interesting findings can be discovered during this process.

» Thanks for the comments. We have added the comparisons of the mass concentration and chemical composition data with previous literatures around the Taiwan Strait and East China Sea per request. The sentences have been revised as "Table 6 compares the concentrations of TC, OC, and EC in PM2.5 with previous studies. The total carbon concentrations were close to those at the Penghu site located at an offshore island where clean marine air can dilute PM2.5 from long-range transport, resulting in local emission accumulation and lower OC and EC levels at the Penghu Islands. The OC/EC ratios ranged from 3.0-7.0 on the southeastern coastline of China and from 1.9 to 2.9 on the southwestern coastline of the Taiwan Strait, respectively. The OC/EC ratios obtained from this study ranged from 2.6 to 2.8 at the Taiwan Strait which were generally lower than those reported by Chou et al., 2010 (2.6 to 2.9) and Tasi et al., 2010 (1.9 to 2.9). The comparison of OC/EC ratios showed the variation of carbonaceous species analyzed with different analytical methods. The carbonaceous concentrations of particulate matter analyzed using thermal optical reflectance (TOR) and thermal optical transmittance (TOT) were generally higher than those using elemental analysis (EA)." (see P14, L12-L20). B. Logic B1: The major discussion including seven sections, and less comparison with the data over other areas, which makes the reader feel that the author is just loading the data. It is better to find some internal connection between these data and name each section following the findings. Or the author could try 3.1.1 to including some sections into one section. e.g. 3.3, 3.4 and 3.5 are all about the sea salt particles, which can be in one section.

» Thanks for the comments. We have converged and further revised the original Sections 3.3, 3.4 and 3.5 as Section 3.3 "Chemical Characteristics of PM2.5 over Sea and at the Offshore Islands" per request. (see P9, L9-P14, L20)

B2:The outline really need to be reconstructed. I often have this problem with my paper that the closely related information is not discussed until much later in the paper. Please try to discuss, at least briefly, all the relevant information on a topic at one place. Otherwise, some issues sounds like mentioned several times. Language or content can be more condensed.

» Thanks for the comments. We have rearranged the content of the Section "Results and Discussion" in the manuscript. In this Section, we discussed and interpreted the results obtained from this study in the following seven subsections. Subsection 3.1 presents the spatiotemporal variation of PM2.5 concentrations over sea and at the Offshore Islands. Subsection 3.2 introduces the transport routes during three cruise sampling campaign. Subsection 3.3 aggregates the chemical characteristic of marine fine particles during three cruise sampling campaign. Subsection 3.4 describes the distribution and source indicators of PM2.5 during three cruise sampling campaign over sea and at the offshore islands. Subsection 3.5 reconstructs the material balance equation for the gravimetric mass of PM2.5 during three cruise sampling campaign. Subsection 3.6 identifies whether the presence of certain metallic elements in PM2.5 were primar-

ily due to natural or anthropogenic processes during the sampling cruise. Subsection 3.7 compares the chloride deficit of PM2.5 with previous studies, respectively. (see P7, L10-L18).

C. Detailed Comments C1: The weather condition should be also mentioned at first as the sampler number is limited and the reader need some general idea on the background air mass condition. » Thanks for the comments. After checking the records of wind speeds and wind direction in the sampling boat, we found that the prevailing wind came from the northeastern direction. Additionally, the backward trajectories showed that air masses blown from the north generally had higher PM2.5 concentrations than those from the south during the cruise sampling campaign. (see P4, L6-L10).

C2: Page 6 line 30 section 3.2: It should be ammonium poor area that $(NH_4)_2SO_4$ is not favored. $NH_4HSO_4$ is more likely.

» Thanks for the comments. We have recalculated the relationship between $NO_3-$, $SO_4^{2-}$ and $NH_4+$. $NO_3-$, $SO_4^{2-}$ and $NH_4+$ that were associated together in the same particulate system in the likely form of $NH_4NO_3$, and $[NH_4]_2SO_4$ or $NH_4HSO_4$. Particulate phase $NH_4+$ concentrations can be calculated using the stoichiometric ratios of different compounds and compared with the measurements. Nitrate is in the form of $NH_4NO_3$, while sulfate is in the forms of either $(NH_4)_2SO_4$ or $NH_4HSO_4$ which can be estimated by equations (1) and (2).

Previous study indicated that ammonia is known to neutralize sulfuric acid irreversibly, and then nitric acid. Additionally, hydrochloric acid may react with gaseous ammonia to form ammonium chloride aerosols. However, in thermodynamic equilibrium conditions ammonium chloride is reported to be 2-3 times more volatile than ammonium nitrate (Stelson and Seinfeld, 1982) and its formation occurs later. Thus, ammonia is believed to be neutralized firstly by sulfuric acid and forms ammonium sulfate and/or ammonium bisulfate (McMurry et al., 1983; Wang et al., 2005; Du et al., 2010). In this study, we assumed that both sulfate ($SO_4^{2-}$) and bisulfate ($HSO_4-$) could be neutralized by ammonia with various portions (see P.9, L12-L24). Results obtained from the calculation of nitrate showed that the predominant inorganic compounds of PM2.5 were ammonium nitrate (NH4NO3) and ammonium bisulfate (NH4HSO4) (see P9, L26-L27).

C3: Page 8 line 9. the data obtained by equation (1) is in the middle level of these three results, however it is not a reason that "the most accurate method to estimate the sea salt concentrations was equation (1)".

» Thanks for the comment. We have revised the sentence describing the equations of sea salt estimates (see equation (3), P10, L25-L27).

C4: Page 9 line 10-15 about the anthropogenic particle influence, it can be one important topic in this manuscript. Suggest the authors make two major concern: anthropogenic source and sea salts Cl deficit. Besides Section 3.5 title is missing in the manuscript.

» Thanks for the comment. We have rearranged the sections according to other reviewer's suggesting and merged the original Section 3.5 into Section 3.3 in the manuscript per request. Moreover, a new Section 3.6 describes the presence of specific metallic elements in PM2.5 primarily emitted from natural or anthropogenic processes during the sampling cruises. Section 3.7 investigates and compares the chloride deficit of PM2.5 with previous studies. (see P16, L1-P18, L2)

C5: Page 12 line 30, an accurate (OC/EC)pri value used in this study should be mentioned in the manuscript. The discussion of OCEC is really poor.

» Thanks for the comment. We have removed the calculation of SOC and POC according to the suggestion from other reviewers. Moreover, we have also revised the title of the Section 3.3.4 as "Carbonaceous Contents of PM2.5 over Sea and at the Offshore Islands," per request. (see P13, L15-P14, L20)

---

## Author Comment (AC2) · 16 Nov 2016

Review for "Chemical Characteristics of Marine Fine Aerosols over Sea and at Offshore Islands during Three Cruise Sampling Campaigns in the Taiwan Strait- Sea Salts and Anthropogenic Particles," by Li et al. submitted to ACPD This paper presents filter-based PM2.5 composition measurements area over Taiwan strait where is not well characterized and compares with the condition at an offshore island site in Penghu island. This is a high quality data set and an important contribution to the field. I do recognize that this is a data set over a large marine area with different cruises

and as such it is difficult to analyze, but I do not think the analysis at present is as thorough and coherent as it could be. Overall comments: This paper presents almost all the compositions can be measured in PM2.5 in three courses over Taiwan Strait, including the major ions, heavy metals and OCEC. The data quality is good however the overall impression is the authors just showed so much data there and did not find any impressed or interesting findings. I suggest to submit to other popular journals such as Atmospheric Environment or Atmospheric Research, unless the author can provide any interesting findings after further analysis.

Detailed comments: A. The tables and figures A1: Units need to be added to the table 2 3 5.

» Thanks for the comments. We have added the units in Tables 3, 4, and 5 per request. (see P26-P28)

A2: Sample numbers should be added in Table 1. Actually I was confused about the sampling method. Totally how many valid filters were collected in different cruises? If the filters of PM2.5-10 were contaminated, it can be moved to the supplemented materials as there are no important results.

» Thanks for the comments. In this study, only one fine particle sample was collected during each sampling course. The number of PM2.5 samples over sea and at the offshore islands during three cruises sampling campaigns was 4, 5, and 4, respectively (see P4, L11-L12 and Table 3, L27).

A3: Tables can contain large information than the one can be illustrated in the manuscript. The authors do not need to mention every data in the tables, but need to add some comparisons with the data in other literatures to rich the contents. Some interesting findings can be discovered during this process.

» Thanks for the comments. We have added the comparisons of the mass concentration and chemical composition data with previous literatures around the Taiwan Strait

and East China Sea per request. The sentences have been revised as "Table 6 compares the concentrations of TC, OC, and EC in PM2.5 with previous studies. The total carbon concentrations were close to those at the Penghu site located at an offshore island where clean marine air can dilute PM2.5 from long-range transport, resulting in local emission accumulation and lower OC and EC levels at the Penghu Islands. The OC/EC ratios ranged from 3.0-7.0 on the southeastern coastline of China and from 1.9 to 2.9 on the southwestern coastline of the Taiwan Strait, respectively. The OC/EC ratios obtained from this study ranged from 2.6 to 2.8 at the Taiwan Strait which were generally lower than those reported by Chou et al., 2010 (2.6 to 2.9) and Tasi et al., 2010 (1.9 to 2.9). The comparison of OC/EC ratios showed the variation of carbonaceous species analyzed with different analytical methods. The carbonaceous concentrations of particulate matter analyzed using thermal optical reflectance (TOR) and thermal optical transmittance (TOT) were generally higher than those using elemental analysis (EA)." (see P14, L12-L20).

B. Logic B1: The major discussion including seven sections, and less comparison with the data over other areas, which makes the reader feel that the author is just loading the data. It is better to find some internal connection between these data and name each section following the findings. Or the author could try 3.1.1 to including some sections into one section. e.g. 3.3, 3.4 and 3.5 are all about the sea salt particles, which can be in one section.

» Thanks for the comments. We have converged and further revised the original Sections 3.3, 3.4 and 3.5 as Section 3.3 "Chemical Characteristics of PM2.5 over Sea and at the Offshore Islands" per request. (see P9, L9-P14, L20)

B2:The outline really need to be reconstructed. I often have this problem with my paper that the closely related information is not discussed until much later in the paper. Please try to discuss, at least briefly, all the relevant information on a topic at one place. Otherwise, some issues sounds like mentioned several times. Language or content can be more condensed.

» Thanks for the comments. We have rearranged the content of the Section "Results and Discussion" in the manuscript. In this Section, we discussed and interpreted the results obtained from this study in the following seven subsections. Subsection 3.1 presents the spatiotemporal variation of PM2.5 concentrations over sea and at the Offshore Islands. Subsection 3.2 introduces the transport routes during three cruise sampling campaign. Subsection 3.3 aggregates the chemical characteristic of marine fine particles during three cruise sampling campaign. Subsection 3.4 describes the distribution and source indicators of PM2.5 during three cruise sampling campaign over sea and at the offshore islands. Subsection 3.5 reconstructs the material balance equation for the gravimetric mass of PM2.5 during three cruise sampling campaign. Subsection 3.6 identifies whether the presence of certain metallic elements in PM2.5 were primarily due to natural or anthropogenic processes during the sampling cruise. Subsection 3.7 compares the chloride deficit of PM2.5 with previous studies, respectively. (see P7, L10-L18).

C. Detailed Comments C1: The weather condition should be also mentioned at first as the sampler number is limited and the reader need some general idea on the background air mass condition.

» Thanks for the comments. After checking the records of wind speeds and wind direction in the sampling boat, we found that the prevailing wind came from the northeastern direction. Additionally, the backward trajectories showed that air masses blown from the north generally had higher PM2.5 concentrations than those from the south during the cruise sampling campaign. (see P4, L6-L10).

C2: Page 6 line 30 section 3.2: It should be ammonium poor area that (NH4)2SO4 is not favored. NH4HSO4 is more likely.

» Thanks for the comments. We have recalculated the relationship between NO3-, SO42- and NH4+. NO3-, SO42- and NH4+ that were associated together in the same particulate system in the likely form of NH4NO3, and [NH4]2SO4 or NH4HSO4. Par-

ticulate phase NH4+ concentrations can be calculated using the stoichiometric ratios of different compounds and compared with the measurements. Nitrate is in the form of NH4NO3, while sulfate is in the forms of either (NH4)2SO4 or NH4HSO4 which can be estimated by equations (1) and (2).

Previous study indicated that ammonia is known to neutralize sulfuric acid irreversibly, and then nitric acid. Additionally, hydrochloric acid may react with gaseous ammonia to form ammonium chloride aerosols. However, in thermodynamic equilibrium conditions ammonium chloride is reported to be 2-3 times more volatile than ammonium nitrate (Stelson and Seinfeld, 1982) and its formation occurs later. Thus, ammonia is believed to be neutralized firstly by sulfuric acid and forms ammonium sulfate and/or ammonium bisulfate (McMurry et al., 1983; Wang et al., 2005; Du et al., 2010). In this study, we assumed that both sulfate (SO42-) and bisulfate (HSO4-) could be neutralized by ammonia with various portions (see P.9, L12-L24). Results obtained from the calculation of nitrate showed that the predominant inorganic compounds of PM2.5 were ammonium nitrate (NH4NO3) and ammonium bisulfate (NH4HSO4) (see P9, L26-L27).

C3: Page 8 line 9. the data obtained by equation (1) is in the middle level of these three results, however it is not a reason that "the most accurate method to estimate the sea salt concentrations was equation (1)".

» Thanks for the comment. We have revised the sentence describing the equations of sea salt estimates (see equation (3), P10, L25-L27).

C4: Page 9 line 10-15 about the anthropogenic particle influence, it can be one important topic in this manuscript. Suggest the authors make two major concern: anthropogenic source and sea salts Cl deficit. Besides Section 3.5 title is missing in the manuscript.

» Thanks for the comment. We have rearranged the sections according to other reviewer's suggesting and merged the original Section 3.5 into Section 3.3 in the manuscript per request. Moreover, a new Section 3.6 describes the presence of specific metallic elements in PM2.5 primarily emitted from natural or anthropogenic processes during the sampling cruises. Section 3.7 investigates and compares the chloride deficit of PM2.5 with previous studies. (see P16, L1-P18, L2)

C5: Page 12 line 30, an accurate (OC/EC)pri value used in this study should be mentioned in the manuscript. The discussion of OCEC is really poor.

» Thanks for the comment. We have removed the calculation of SOC and POC according to the suggestion from other reviewers. Moreover, we have also revised the title of the Section 3.3.4 as "Carbonaceous Contents of PM2.5 over Sea and at the Offshore Islands," per request. (see P13, L15-P14, L20)

Please also note the supplement to this comment:
http://www.atmos-chem-phys-discuss.net/acp-2016-384/acp-2016-384-AC2-supplement.pdf

**Supplement:**

[revised manuscript text omitted]
. Quartz filters were selected in this study since we are interested in the chemical composition of water-soluble ionic species, metallic elements, and carbonaceous content. Before weighing, the quartz-fiber filters were equilibrated in a desiccator at temperatures between 20°C and 25°C and relative humidities (RH) between 35% and 45% for forty-eight hours. After conditioning, the filters were then weighed by a microbalance with the precision of 1 μg to determine the $PM_{2.5}$ mass. The moisture could be mostly removed in the process of conditioning. *(Cheng and Tsai, 2000; Yuan et al., 2006)* (Lines 169-176) Previous studies reported that the 24-h mass concentration differences between quartz-fiber fiber and Teflon filters has no significant difference (*Appel et al., 1984; Tsai et al., 2012*).

[revised manuscript text omitted]

15 The water-soluble ionic species of $PM_{2.5}$ were analyzed by IC, while the metallic contents were analyzed by ICP in this study. This study aims to characterize the chemical composition (especially water-soluble ionic species) of marine fine particles ($PM_{2.5}$) in the atmosphere during the highly polluted seasons and to investigate how importance of sea salt and anthropogenic particles influenced $PM_{2.5}$ in the Taiwan Strait. Therefore, the discussion of chemical species in $PM_{2.5}$ focused on water-soluble ionic species in the manuscript. Furthermore, the comparison of metallic content in $PM_{2.5}$ focused on

20 anthropogenic sources from their original region in the Taiwan Strait. A scatter diagram has been plotted in Figure 2 shown below per request. It showed that the concentrations of metallic contents (Mg, K, and Ca) analyzed by ICP-AES were always higher than those of water-soluble ionic species analyzed by IC.

The carbonaceous contents (i.e. elemental, organic, and total carbons) of $PM_{2.5}$ were measured with an elemental analyzer (EA) (Carlo Erba, Model 1108). Prior to sampling, the quartz fiber filters had to be pre-heated at 900 °C for 1.5 h to expel

25 the potential organic impurities. This preheating procedure minimized the background carbonaceous species in the quartz fiber filters and matrix, which would interfere with the analytical results, possibly leading to an overestimation of the carbonaceous species of $PM_{2.5}$. The elemental analyzer was operated using the procedure of oxidation at 1020°C and that of reduction at 500°C, for continuous 15-min heating. Additionally, one eighth of the quartz fiber filter was heated in advance by hot nitrogen gas (340-345°C) for 30 min to expel the organic carbon (OC) fraction, after which the amount of elemental

30 carbon (EC) was determined. Another eighth of the quartz fiber filter was analyzed without heating, and the carbonaceous species thus characterized as total carbon (TC). The amount of organic carbon (OC) could be determined by subtracting the elemental carbon from total carbon. Although the aforementioned thermal analysis was the most widely used method for determining the carbonaceous species in marine $PM_{2.5}$ aerosols, a charring formation error from filter preheating was not

taken into account for correction, and this artifact might cause the overestimation of EC and the underestimation of OC (*Li et al., 2013a; 2013b ;2015; 2016*).

**2.3 Quality Assurance and Quality Control**

Quality assurance and quality control (QA/QC) for both sampling and chemical analysis of $PM_{2.5}$ were also employed in this
5    study. Prior to sampling, the flow rate of each $PM_{2.5}$ sampler was carefully calibrated with an orifice calibrator (SENSIDYNE, MCH-01). The high-volume air sampler (TE-6001) was used to collect $PM_{2.5}$ with a sampling flow rate of 1.47 $m^3$/min passing through a $PM_{2.5}$ selective inlet. As the particulates travel through the $PM_{10}$ size selective inlet the larger particulates are trapped inside of the inlet as the smaller $PM_{2.5}$ particulates continue to travel through the inlet and are collected on the 8" x 10" quartz fiber filter manufactured by Pall Corporation. This method was complied with the sampling
10   method of NIEA A102.12A similar to USEPA Method IO-2.1. Quartz fiber filter was selected for this study because we conducted the chemical analysis of water-soluble ions, metallic elements, and carbonaceous content. Before weighing, the quartz fiber filters were conditioned in a desiccator at temperatures of 20-25°C and relative humidity (RH) of 35-45% for 48 hours. After conditioning, the filters were then weighed by a microbalance (Sartorius MC 5) with the precision of 1 μg to determine the mass of $PM_{2.5}$. The filters were stored in a weighing chamber at temperatures of 20-25°C and relative humidity
15   of 35-45%. The quartz fiber filters were handled with care, so as to prevent potential contamination and cracking during the sampling procedure, as they were placed on the $PM_{2.5}$ samplers. After sampling, aluminium foil was used to fold the quartz fiber filters, which were then temporarily stored at $4^o$C and transported back to the laboratory for chemical analysis within two days. The sampling and analytical procedures were similar to those described in previous studies (*Cheng and Tsai, 2000; Lin, 2002; Yuan et al., 2006; Tsai et al., 2008; Tsai et al., 2010; Witz et al., 1990*). In the sampler, an adapter is placed into
20   the model TE-6001 sampler in lieu of an existing $PM_{10}$ fractionator. The adapter has a plate that contains multiple impactors for collecting particles larger than $PM_{2.5}$ on a slotted quartz fiber filter. $PM_{2.5}$ is then passed through the impactor and collected on a quartz fiber filter. After sampling, the concentrations of $PM_{2.5}$ and $PM_{2.5-10}$ were determined by weighting the quartz fiber filter and the slotted quartz fiber filter, respectively. Due to the difficulty of identically dividing the slotted quartz fiber filter, we thus solely analyzed the chemical compositions of $PM_{2.5}$.
25   Both field and transportation blanks were further undertaken for $PM_{2.5}$ sampling, while reagent and filter blanks were applied for chemical analysis. Summary of QA/QC results for the analysis of chemical species in this study are summarized in Table 1. The determination coefficient ($R^2$) of the calibration curve for each chemical species was required to be higher than 0.995. Background contamination was routinely monitored by using operational blanks (unexposed filters); these were processed simultaneously with the field samples. The present experimental works found that the background interference was
30   insignificant and could thus be ignored in this study. At least 10% of the samples were analyzed by spiking with a known amount of metallic elements and water-soluble ionic species to determine their recovery efficiencies. The results of the

recovery efficiency tests indicated that the recovery efficiencies among every 10 filter samples varied from 96 to 103%. In addition, the results of the reproducibility varied from 97 to 104% for all the chemical species.

**2.4 Transport Routes of Air Masses**

In order to identify the predominant sources of air pollutants, backward trajectory has been widely used to trace the transport routes of air masses (Li et al., 2016). The Hybrid Single-Particle Lagrangian Integrated Trajectory (HYSPLIT) is a widely used model that plots the trajectory of a single air parcel from a specific location and height above ground over a period of time. The 72-hr backward trajectories started at the island site (23°33'41"N; 119°35'10"E) at the altitudes of 100, 350, and 500 m above sea level, respectively.

**3 Results and discussion**

In this Section, we discussed and interpreted the results obtained from this study in the following seven subsections. Subsection 3.1 presents the spatiotemporal variation of $PM_{2.5}$ concentrations over sea and at the Offshore Islands. Subsection 3.2 introduces the transport routes during three cruise sampling campaign. Subsection 3.3 aggregates the chemical characteristic of marine fine particles during three cruise sampling campaign. Subsection 3.4 describes the distribution and source indicators of $PM_{2.5}$ during three cruise sampling campaign over sea and at the offshore islands. Subsection 3.5 reconstructs the material balance equation for the gravimetric mass of $PM_{2.5}$ during three cruise sampling campaign. Subsection 3.6 identifies whether the presence of certain metallic elements in $PM_{2.5}$ were primarily due to natural or anthropogenic processes during the sampling cruise. Subsection 3.7 compares the chloride deficit of $PM_{2.5}$ with previous studies, respectively.

**3.1 Spatiotemporal Variation of $PM_{2.5}$ Concentrations over Sea and at the Offshore Islands**

The mass concentrations of $PM_{2.5}$ and $PM_{2.5-10}$, and the mass ratios of $PM_{2.5}/PM_{10}$ over sea and at the offshore islands during the $PM_{2.5}$ sampling campaigns are summarized in Table 2. Limited to the voyage of air quality monitoring boat, each sampling course was arranged to collect $PM_{2.5}$ for only continuous 8-12 hours (i.e. daytime and nighttime) based on the distance of navigation during the sampling cruise over sea. On the islands, $PM_{2.5}$ was sampled for continuous 24 hours. Field sampling results indicated that the concentrations of $PM_{2.5}$ were generally higher than those of $PM_{2.5-10}$ with the exception of the winter cruise in 2014 (W14).

The mass ratios of $PM_{2.5}/PM_{10}$ over sea and at the offshore islands ranged from 42.4% to 44.0% for the winter cruise of 2014 (W14), which were lower than those for other two sampling cruises ranging from 49.2% to 55.7%. The concentrations of $PM_{2.5-10}$ over sea were always higher than those at the offshore islands, indicating that sea salt highly influenced the marine aerosols with a dominant size range of 2.5-10 μm. Higher $PM_{2.5}$ concentrations were commonly observed over sea. The fine

particles might be emitted from vessels including commercial ships and fishery boats were significant contributions to marine fine aerosols and cannot be ignored over the seas and oceans where the marine transportation traffics are busy. Unexpectedly, a similar trend was also observed for $PM_{2.5}$.with an exception of winter cruise of 2013 (W13). There are two possible suggestions or explanations for this phenomenon. One possibility suggested that fine particles emitted from vessels including commercial ships and fishery boats were significant contributions to marine fine aerosols and cannot be ignored over the seas and oceans where the marine transportation traffics are busy. The other possibility indicated that long-range cross-boundary transportation of fine particles emitted from the upwind regions such as China, Korea, and/or Japan could be constantly transported to the target areas and even across the Taiwan Strait.

Moreover, higher $PM_{2.5}$ concentrations were commonly observed in the daytime than those at nighttime for all three cruise sampling campaigns. It was mainly attributed to the facts that higher frequency of human activities such as commercial burning, vehicle and vessel traffics in the daytime than at nighttime. Additionally, the sea-land breezes play an important role in transporting air pollutants to and from urban areas on the coastal regions *(Ding et al., 2004; Tsai et al., 2008)*. Previous study stated that the surface wind fields at the Penghu Islands varied in the daytime and at nighttime (Li et al., 2015). In the daytime, air masses were divided into two separate zones. On the west side of the Penghu Islands, a strong northward prevailing wind was coupled with the sea-land breeze, causing by the sea breezes in the southwestern coastal region of the Taiwan Island. Conversely, the surface wind directions moved to the east on the east-side of the Penghu Islands. *Viana et al. (2005)* also stated that atmospheric coastal dynamics exert a significant influence on the levels and composition of atmospheric particles. Sea breezes may penetrate deep inland and cause high pollutant episodes in early afternoon, resulting in higher concentration of marine aerosols in the daytime than that at nighttime *(Ding et al., 2004)*.

**3.2 Transport Routes of $PM_{2.5}$ Concentrations over Sea and at the Offshore Islands**

Previous study reported that the level of atmospheric $PM_{2.5}$ is affected by meteorological condition, thus $PM_{2.5}$ concentrations in spring and winter was much higher than those in fall and summer in the Taiwan Strait (Li et al., 2016b). However, the results of backward trajectory analysis were not shown in this manuscript. Our previous study indicated that the corresponding trajectories were clustered into three major transport routes according to their airflow directions and regions through which air masses traveled toward the Taiwan Strait (Li et al., 2016b). During the consecutive courses in the winter of 2013 (W3), air masses originated from Mongolia were transported across the northern, central, and southeastern China. During the consecutive courses in the spring of 2014 (S14), air masses originated from northern and northeastern China were transported through the coastal regions of central China, East China Sea, and southeastern China toward the Taiwan Strait. During the consecutive courses in the winter of 2014 (W14), air masses originated from northern and northeastern China are transported through the coastal regions of central and southeastern China toward the Taiwan Strait. Results from backward trajectories showed that the concentrations of $PM_{2.5}$ blown from the north were generally higher than

those from the south. PM$_{2.5}$ samples were collected over sea and at the offshore site during the high pollution seasons in this study.

In this study, 72-hour backward trajectories ending at the Penghu Islands at the altitudes of 100, 350, and 500 m above sea level, respectively, were simulated to represent air masses toward the Taiwan Strait. Air masses originated from Mongolia were transported through northern and central China, and finally across the East China Sea to the offshore site during the three sampling cruises. The results indicated that anthropogenic chemical species were evenly dispersed over sea for the same trajectory during the air pollution episodes, causing the carbonaceous species stably distributed over sea and at the offshore site.

**3.3 Chemical Characteristics of PM$_{2.5}$ over Sea and at the Offshore Islands.**

**3.3.1 Water-soluble Ions of PM$_{2.5}$**

The chemical compositions of PM$_{2.5}$ sampled over sea and at the offshore islands in the Taiwan Strait during the cruise sampling campaigns are summarized in Table 3. $NO_3^-$, $SO_4^{2-}$ and $NH_4^+$ that were associated together in the same particulate system in the likely form of $NH_4NO_3$, and $[NH_4]_2SO_4$ or $NH_4HSO_4$. Particulate phase $NH_4^+$ concentrations can be calculated using the stoichiometric ratios of different compounds and compared with the measurements. Nitrate is in the form of $NH_4NO_3$, while sulfate is in the forms of either $(NH_4)_2SO_4$ or $NH_4HSO_4$ which can be estimated by equations (1) and (2).

$$(NH_4)_2SO_4 = 0.38 \times SO_4^{2-} + 0.29 \times NO_3^- \tag{1}$$

$$NH_4HSO_4 = 0.192 \times SO_4^{2-} + 0.29 \times NO_3^- \tag{2}$$

Previous study indicated that ammonia is known to neutralize sulfuric acid irreversibly, and then nitric acid. Additionally, hydrochloric acid may react with gaseous ammonia to form ammonium chloride aerosols. However, in thermodynamic equilibrium conditions ammonium chloride is reported to be 2-3 times more volatile than ammonium nitrate *(Stelson and Seinfeld, 1982)* and its formation occurs later. Thus, ammonia is believed to be neutralized firstly by sulfuric acid and forms ammonium sulfate and/or ammonium bisulfate *(McMurry et al., 1983; Wang et al., 2005; Du et al., 2010)*. In this study, we assumed that both sulfate ($SO_4^{2-}$) and bisulfate ($HSO_4^-$) could be neutralized by ammonia with various portions. The most abundant water-soluble ionic species of PM$_{2.5}$ were $NO_3^-$, $SO_4^{2-}$, $NH_4^+$, Cl$^-$, and Na$^+$ indicating that secondary inorganic aerosols (SIAs) and sea salt was the major portion of PM$_{2.5}$. Results obtained from the calculation of nitrate showed that the predominant inorganic compounds of PM$_{2.5}$ were ammonium nitrate ($NH_4NO_3$) and ammonium bisulfate ($NH_4HSO_4$), which was formed by the neutralization of sulfuric and nitric acids with ammonia. The ionic balance (i.e. A/C) of ionic species ranged from 0.7 to1.0, including that PM$_{2.5}$ were acidic particles. The results indicated that both sea salt 
[revised manuscript text omitted]

Figure 5 illustrates the spatiotemporal variation of carbonaceous contents and their mass ratios (OC/EC) of $PM_{2.5}$ sampled over sea and at the offshore islands during three cruise sampling campaigns. Carbonaceous compounds in the atmosphere

25   consist mainly of elemental and organic carbons (EC and OC). EC is a product of incomplete combustion from residential coals, motor vehicle fuels, and biomass burning. OC originates from primary anthropogenic sources like above mentioned combustions and from the formation of secondary OC by chemical reactions in the atmosphere. The concentrations of OC in $PM_{2.5}$ were always higher than those of EC at the offshore site and over sea around the Taiwan Strait, while the OC/EC ratio ranged from 2.2 to 2.7, respectively. The concentrations of OC were mostly higher at the offshore site than those over sea

30   during the sampling periods. The OC/EC ratios ranged from 2.3 to 2.6 at the offshore site, and from 2.2 to 2.6 at the south Taiwan Strait (routes of W13C1, S14C1, and W14C1), 2.2 to 2.7 at the central Taiwan Strait adjacent to Penghu Islands (routes of W13C2, S14C2, and W14C2), 2.2 to 2.5 at the routes which were adjacent to coastline of western Taiwan Islands

of returning to Kaohsiung Port (W13C3, S14C4, and W14C3), and accounted for 2.6 at the northern Taiwan Strait (S14C3), respectively. The higher OC/EC ratios were generally observed at the central and northern Taiwan Strait, and follow by the coastline of western Taiwan Islands and southern Taiwan Strait. Previous studies indicated that the OC/EC ratio has been used in many previous studies to determine whether the carbonaceous aerosols are primary or secondary. A high OC/EC

5   ratio coupled with a poor correlation implies an influx of urban pollutants from elsewhere or the formation of secondary OC (SOC) from photochemical reactions. On the other hand, a high correlation indicates primary emission and a secondary formation derived from the primary carbon (*Turpin et al., 1991; Strader et al., 1999*). Previous reports also appointed that high OC/EC ratio in fall and winter might be resulted from (1) more coal burning for space heating in winter at the North China; (2) more conversion of semi-volatile organic materials into particles in lower temperatures; and (3) more secondary

10  organic carbon production as air masses passed from industrial and highly pollutant areas under specific meteorological conditions in fall and winter (*Duan et al., 2007; Li et al., 2016*).

Table 6 compares the concentrations of TC, OC, and EC in $PM_{2.5}$ with previous studies. The total carbon concentrations were close to those at the Penghu site located at an offshore island where clean marine air can dilute $PM_{2.5}$ from long-range transport, resulting in local emission accumulation and lower OC and EC levels at the Penghu Islands. The OC/EC ratios

15  ranged from 3.0-7.0 on the southeastern coastline of China and from 1.9 to 2.9 on the southwestern coastline of the Taiwan Strait, respectively. The OC/EC ratios obtained from this study ranged from 2.6 to 2.8 at the Taiwan Strait which were generally lower than those reported by *Chou et al., 2010* (2.6 to 2.9) and *Tasi et al., 2010* (1.9 to 2.9). The comparison of OC/EC ratios showed the variation of carbonaceous species analyzed with different analytical methods. The carbonaceous concentrations of particulate matter analyzed using thermal optical reflectance (TOR) and thermal optical transmittance

20  (TOT) were generally higher than those using elemental analysis (EA).

**3.4 Distribution and Source Indicators of $PM_{2.5}$ over Sea and at the Offshore Islands**

The results of the distribution percentage of chemical characteristics to $PM_{2.5}$ during the three courses over Taiwan Strait indicated that water-soluble ionic species, metallic elements, and carbonaceous content accounted for 46.1-52.0%, 14.7-

25  19.3%, and 14.0-19.9% of $PM_{2.5}$ during the sampling courses in the winter of 2013, respectively; 44.4-54.1%, 13.1-15.7%, and 13.3-17.9% during the sampling courses in the spring of 2014, respectively; 42.7-45.8%, 12.3-13.4%, and 15.2-18.8% during the sampling courses in the winter of 2014, respectively. The results indicated that the distribution of water-soluble ionic species and carbonaceous contents on the offshore islands were generally higher than those over sea. Previous study reported that the emissions of huge amounts of particulates from various sources (e.g., textile plants at the Jinjing River

30  Basin) could result in the higher percentages of ionic and carbonaceous contents at the Taiwan Strait (Li et al., 2016a). The results were close to those at the Penghu site located at an offshore island.

There are several ratios of chemical species can be used as valuable indicators to appoint atmospheric particles from specific sources (Cao et al., 2012; Arimoto et al., 1992). Previous researches reported that the mass ratios of EC/TC, $K^+$/TC, and TC/$SO_4^{2-}$ can be used to identify the sources from biomass burning (VanCuren, 2013). When the mass ratio of EC/TC ranges from 0.1 to 0.2, $K^+$/TC ranges from 0.5 to 1.0, and TC/$SO_4^{2-}$ ranges from 6 to 15, it suggests that the sources were mainly contributed from biomass burning (VanCuren, 2013). The mass ratios of $NO_3^-$/nss-$SO_4^{2-}$ have also been used to evaluate the contributions from stationary and mobile sources (Arimoto et al., 1992). The mass ratios of $NO_3^-$/nss-$SO_4^{2-}$ higher than unity indicated that the sources of particles were mainly from mobile sources. Conversely, the mass ratios of $NO_3^-$/nss-$SO_4^{2-}$ lower than unity suggested that the sources of particles came mainly from stationary sources. Table 7 compares the mass ratios of major chemical species over sea and at the offshore islands. The mass ratios of EC/TC, $K^+$/TC, $NO_3^-$/nss-$SO_4^{2-}$, and TC/$SO_4^{2-}$ during the three courses over Taiwan Strait ranged from 0.28±0.01 to 0.30±0.02 in the winter of 2013 (W13), 0.10±.02 to 0.17±.05, 0.66±0.05 to 0.71±0.03 in the spring of 2014 (S14), and 0.94±0.05 to 1.18±0.08 in the winter of 2014 (W14), respectively, while similar trends reported at the southeastern coastline of the Taiwan Strait (Li et al., 2016a). Previous studies reported that high $SO_4^{2-}$ and $NO_3^-$ concentrations observed at the Penghu site and the Kaohsiung site were mainly from stationary sources due to burgeoning industrial development in the southwestern coastal region of Taiwan (Tsai et al., 2011; Li et al., 2016a). According to the reports from *VanCuren (2003)*, atmospheric aerosols with high nss-$SO_4^{2-}$/$NO_3^-$ ratios were attributed mainly from stationary sources.

**3.5 Reconstruction of $PM_{2.5}$ over Sea and at the Offshore Islands**

$PM_{2.5}$ was estimated by material balance equation for gravimetric mass (Chow et al., 1996). In this study, $PM_{2.5}$ was summed by nine major chemical compositions: nitrate ($NO_3^-$), sulfate ($SO_4^{2-}$), ammonium ($NH_4^+$), chloride ($Cl^-$), organic matter (OM), elemental carbon (EC), crustal materials (CM), sea salt, and others. Organic material (OM) was estimated from an organic carbon (OC) multiplier (f) that accounts for unmeasured hydrogen (H), oxygen (O), nitrogen (N), and sulfur (S) in organic compounds (Chow et al., 1996). Multipliers of 1.4 to 1.8 have been found to best represent the complex mixture of organic molecules in OM (POM) (Chow et al., 1996). A factor of 1.6 for converting OC to OM was used in this study. Crustal materials can be estimated using a method reported by Wedepohl (1995) (Crustal materials= 12*[Al]), while sea salt was estimated by equation (3). Therefore, $PM_{2.5}$ concentrations were reconstructed by the following equation, (8),

$$[PM_{2.5}] = SO_4^{2-} + NO_3^- + NH_4^+ + (Organic.Materials) + (Elemental.Carbon) \\ + (Crustal.Materials) + (Sea.Salt) + (Trace.Element) + (Others) \tag{8}$$

Table 8 compares the major chemical components of $PM_{2.5}$ at the coastal sites around the Taiwan Strait and East China Sea. Consistent with previous studies, organic materials (OM), $SO_4^{2-}$, $NO_3^-$, and crustal materials (CM) were the major components for the reconstruction of $PM_{2.5}$ concentration. The results indicated that $SO_4^{2-}$, $NO_3^-$, crustal materials, and organic materials are important components in $PM_{2.5}$ at all coastal sites. The contribution of $SO_4^{2-}$, $NO_3^-$, and organic materials were similar to other coastal sites.

**3.6 Enrichment Factors of $PM_{2.5}$ over Sea and at the Offshore Islands**

To identify whether the presence of certain metallic elements in aerosols were primarily due to natural or anthropogenic processes, the enrichment factor (EF) of each metallic element was further determined in this study. In previous studies, the particle-induced X-ray emission or proton-induced X-ray emission (PIXE) was used to determine the elemental characteristics of atmospheric particles (Artaxo et al. 1992). In this study, we analyzed ionic species and metallic content by IC and ICP-AES. Comparing different methods for analyzing chemical composition is difficult to calculate the enrichment factors of ionic species (e.g. $Cl^-$), respectively. In this study, same methods were applied to analyse $PM_{2.5}$ samples collected at the target region, which should be comparable for the enrichment factors of $PM_{2.5}$ over sea and at the offshore islands. Thus, we used the presence of certain metallic elements in aerosols surrounding the Taiwan Strait primarily due to natural or anthropogenic processes in the target regions. Although Al, Fe, and Ca are prominent crustal elements which have been used as the reference elements in previous literatures, aluminum (Al) was used as the reference element for the EF calculations for this particular study. The non-crustal EF can be estimated by equation (9) (*Chester et al., 2000*).

$$EF = \frac{(C_x / Al)_{aerosol}}{(C_x / Al)_{crust}} \tag{9}$$

where $(C_x/Al)_{aerosol}$ represents the concentration ratio of specific metallic element ($C_x$) to Al in aerosols; and $(C_x/Al)_{crust}$ represents the corresponding ratio of specific metallic element ($C_x$) to Al in the crustal matter.

The EF values of metallic elements in $PM_{2.5}$ over sea and at the offshore islands are depicted in Figure 6. The order of the EF values for various metallic elements had quite similar trend no matter where atmospheric $PM_{2.5}$ were sampled. For least ten measured metallic elements, their EF values were in the range of 0.1 to 10000 and highly relevant. Trace elements Ni and Cr were highly enriched (100<EF<10000) in $PM_{2.5}$, while Mn and Pb were moderately enriched (10<EF<100) at all sites around the Taiwan Strait. Previous studies reported that metallic elements with EF>10 have an important proportion of non-crustal sources and that a variety of emission sources could contribute to their loading in the ambient air. The EF values of crustal elements Mg, K, Ca, and Fe in $PM_{2.5}$ ranged from 1 to 10 over sea and at the offshore islands, and their EF values were quite consistent for fine particles sampled at different sites. It suggested that these crustal elements were likely originated from same natural sources and had no enrichment in $PM_{2.5}$. In comparison, high EF values of Ni, Cr, Mn, and Pb in the range of 10-10000 suggested that these trace elements were mainly originated from anthropogenic sources. Previous study reported that metallic elements Cr and Ni in $PM_{2.5}$ were mainly from anthropogenic combustion sources, while Cr and Ni in $PM_{2.5-10}$ had more soil-related origins (*Chow et al., 1995*).

[revised manuscript text omitted]

$R^2$: determination coefficient calibration curve;MDL: method detection limit;RPD: relative percentage difference; Units of MDL and Blank: $\mu g\ m^{-3}$

**Table 2. The mass concentrations of PM$_{2.5}$ and PM$_{2.5\text{-}10}$, and the mass ratios of PM$_{2.5}$/PM$_{10}$ over sea and at the offshore islands during three cruise sampling campaigns.**

| Sampling Sites | Sampling Abbr. | Day/Night Samples | Sampling Duration | Sampling Periods | PM$_{2.5}$ (μg m$^{-3}$) | PM$_{2.5\sim10}$ (μg m$^{-3}$) | PM$_{10}$ (μg m$^{-3}$) | PM$_{2.5}$/PM$_{10}$ (%) | Reference |
|---|---|---|---|---|---|---|---|---|---|
| On the Island | W13 | Daily | 24 hrs | Winter 2013 | 28.2 | 23.3 | 51.5 | 54.8 | |
| | S14 | Daily | 24 hrs | Spring 2014 | 22.7 | 19.7 | 42.4 | 53.5 | |
| | W14 | Daily | 24 hrs | Winter 2014 | 38.1 | 30.3 | 68.3 | 55.7 | |
| Over Sea | W13C1 | Nighttime | 8 hrs | Winter 2013 | 21.5 | 22.2 | 43.7 | 49.2 | This study |
| | W13C2 | Daytime | 12 hrs | | 27.0 | 22.2 | 49.2 | 55.0 | |
| | W13C3 | Nighttime | 12 hrs | | 26.0 | 23.4 | 49.3 | 52.6 | |
| | S14C1 | Nighttime | 12 hrs | Spring 2014 | 30.2 | 29.3 | 59.5 | 50.8 | |
| | S14C2 | Daytime | 12 hrs | | 31.7 | 31.1 | 62.7 | 50.5 | |
| | S14C3 | Nighttime | 12 hrs | | 31.3 | 30.5 | 61.8 | 50.7 | |
| | S14C4 | Daytime | 16 hrs | | 33.4 | 31.2 | 64.6 | 51.7 | |
| | W14C1 | Nighttime | 8 hrs | Winter 2014 | 38.6 | 52.5 | 91.1 | 42.4 | |
| | W14C2 | Daytime | 12 hrs | | 43.5 | 55.3 | 98.8 | 44.0 | |
| | W14C3 | Nighttime | 12 hrs | | 38.3 | 51.0 | 89.3 | 42.9 | |
| West-side of the Taiwan Strait | XM,FZ | Daily | 24 hrs | Winter 2013 | 67.4 | 54.7 | 122.0 | 55.2 | Li et al., 2016a |
| | | | | Spring 2014 | 78.6 | 51.0 | 129.6 | 60.6 | |
| | | | | Winter 2014 | 85.8 | 75.0 | 160.8 | 50.3 | |
| East-side of the Taiwan Strait | KH, TC, TP | Daily | 24 hrs | Winter 2013 | 48.0 | 22.98 | 71.0 | 67.6 | |
| | | | | Spring 2014 | 33.8 | 28.5 | 62.3 | 54.3 | |
| | | | | Winter 2014 | 53.4 | 54.6 | 108.0 | 49.5 | |

Islands represents the Xiaomen site at the Penghu Islands.

**Table 3. Chemical composition of PM$_{2.5}$ sampling over sea and at the offshore islands during three cruise sampling campaigns.**

| Cruise Campaigns | Winter 2013 (W13) (n=4) | | | | Spring 2014 (S14) (n=5) | | | | | Winter 2014 (W14) (n=4) | | | |
|---|---|---|---|---|---|---|---|---|---|---|---|---|---|
| Sampling Sites | Islands | Over Sea | | | Islands | Over Sea | | | | Islands | Over Sea | | |
| Species | W13 | W13C1 | W13C2 | W13C3 | S14 | S14C1 | S14C2 | S14C3 | S14C4 | W14 | W14C1 | W14C2 | W14C3 |
| PM$_{2.5}$ | 28.2 | 21.5 | 27.0 | 26.0 | 22.7 | 30.2 | 31.7 | 31.3 | 33.4 | 37.6 | 38.6 | 43.5 | 38.3 |
| SO$_4^{2-}$ | 4.8 | 3.6 | 4.1 | 4.2 | 4.3 | 4.9 | 4.4 | 4.6 | 5.0 | 6.5 | 5.2 | 5.6 | 5.3 |
| NO$_3^-$ | 3.1 | 2.5 | 2.6 | 2.9 | 2.3 | 3.0 | 2.9 | 2.9 | 3.2 | 4.0 | 3.2 | 3.4 | 3.4 |
| NH$_4^+$ | 1.2 | 0.8 | 1.0 | 1.2 | 1.1 | 2.0 | 1.9 | 1.9 | 2.1 | 2.2 | 2.4 | 2.6 | 2.3 |
| Cl$^-$ | 1.8 | 1.7 | 1.9 | 2.0 | 1.5 | 2.2 | 2.1 | 2.1 | 2.2 | 1.4 | 2.8 | 2.8 | 2.9 |
| Na$^+$ | 1.2 | 1.2 | 1.3 | 1.3 | 1.0 | 1.5 | 1.3 | 1.5 | 1.4 | 1.0 | 1.9 | 2.0 | 1.9 |
| K$^+$ | 1.5 | 0.5 | 0.7 | 0.5 | 1.1 | 0.4 | 0.5 | 0.3 | 0.5 | 0.9 | 0.5 | 0.7 | 0.6 |
| Mg$^{2+}$ | 0.5 | 0.6 | 0.6 | 0.6 | 0.5 | 0.7 | 0.7 | 0.6 | 0.6 | 0.6 | 0.8 | 0.8 | 0.6 |
| Ca$^{2+}$ | 0.3 | 0.3 | 0.4 | 0.3 | 0.5 | 0.4 | 0.4 | 0.4 | 0.5 | 0.6 | 0.5 | 0.6 | 0.6 |
| A/C | 0.9 | 0.9 | 0.9 | 1.0 | 0.8 | 0.8 | 0.8 | 0.8 | 0.8 | 0.9 | 0.7 | 0.7 | 0.8 |
| Mg | 0.8 | 0.7 | 0.7 | 0.8 | 0.6 | 0.7 | 0.8 | 0.7 | 0.8 | 0.6 | 0.8 | 0.9 | 0.8 |
| K | 1.7 | 0.7 | 0.8 | 0.8 | 1.5 | 0.6 | 0.6 | 0.6 | 0.7 | 1.2 | 0.8 | 0.9 | 0.7 |
| Ca | 0.4 | 0.7 | 0.6 | 0.6 | 0.6 | 0.8 | 0.8 | 0.7 | 0.8 | 0.7 | 0.6 | 0.7 | 0.7 |
| Cr | 0.1 | 0.0 | 0.1 | 0.2 | 0.2 | 0.1 | 0.1 | 0.2 | 0.2 | 0.3 | 0.2 | 0.3 | 0.3 |
| Mn | 0.2 | 0.2 | 0.2 | 0.3 | 0.2 | 0.2 | 0.3 | 0.3 | 0.3 | 0.2 | 0.2 | 0.2 | 0.2 |
| Fe | 0.4 | 0.1 | 0.4 | 0.2 | 0.4 | 0.2 | 0.3 | 0.2 | 0.3 | 0.3 | 0.2 | 0.3 | 0.3 |
| Zn | 0.2 | 0.1 | 0.1 | 0.2 | 0.1 | 0.1 | 0.1 | 0.1 | 0.2 | 0.2 | 0.1 | 0.2 | 0.2 |
| Al | 0.9 | 0.3 | 0.5 | 0.6 | 0.6 | 0.7 | 0.7 | 0.7 | 0.7 | 1.2 | 0.8 | 0.8 | 0.8 |
| Pb | 0.1 | 0.2 | 0.2 | 0.3 | 0.2 | 0.2 | 0.2 | 0.2 | 0.3 | 0.2 | 0.3 | 0.4 | 0.3 |
| Ni | 0.3 | 0.1 | 0.1 | 0.2 | 0.2 | 0.2 | 0.3 | 0.3 | 0.3 | 0.3 | 0.3 | 0.3 | 0.3 |
| OC | 4.1 | 2.3 | 2.9 | 2.5 | 2.8 | 3.1 | 3.3 | 3.0 | 3.2 | 5.1 | 4.2 | 5.2 | 4.5 |
| EC | 1.5 | 1.1 | 1.3 | 1.1 | 1.2 | 1.3 | 1.2 | 1.2 | 1.4 | 2.0 | 1.7 | 2.1 | 1.8 |
| Sea salt | 3.6 | 3.5 | 3.8 | 3.9 | 3.0 | 4.5 | 3.9 | 4.2 | 4.3 | 2.9 | 5.5 | 5.7 | 5.7 |
| Sea salt/PM$_{2.5}$ (%) | 12.7 | 16.1 | 13.9 | 15.0 | 13.2 | 14.7 | 12.4 | 13.4 | 12.8 | 7.7 | 14.3 | 13.1 | 14.8 |
| WSI/PM$_{2.5}$(%) | 50.8 | 52.0 | 46.1 | 49.9 | 54.1 | 49.9 | 44.4 | 45.3 | 46.4 | 45.8 | 44.7 | 42.7 | 45.7 |
| Metal/PM$_{2.5}$(%) | 19.3 | 15.4 | 14.7 | 16.7 | 15.7 | 13.5 | 13.7 | 13.1 | 14.6 | 12.5 | 12.3 | 12.3 | 13.4 |
| Carbon/PM$_{2.5}$(%) | 19.9 | 15.6 | 15.4 | 14.0 | 17.9 | 14.6 | 14.3 | 13.3 | 13.9 | 18.8 | 15.2 | 16.8 | 16.6 |

Unit: μg m$^{-3}$; n: sample numbers; WSI: water-soluble ionic species; Metal: total metallic contents; Carbon: total carbonaceous contents; A/C: the ratio of micro-equivalent concentrations of anions and cations

**Table 4. The concentrations of sea-salt and non-sea salt water-soluble ionic species of $PM_{2.5}$ over sea and at the offshore islands during three cruise sampling campaigns.**

| Cruise Campaigns | Winter 2013 (W13) | | | | Spring 2014 (S14) | | | | | Winter 2014 (W14) | | | |
|---|---|---|---|---|---|---|---|---|---|---|---|---|---|
| Sampling Sites | Islands | Over Sea | | | Islands | Over Sea | | | | Islands | Over Sea | | |
| Species | W13 | W13C1 | W13C2 | W13C3 | S14 | S14C1 | S14C2 | S14C3 | S14C4 | W14 | W14C1 | W14C2 | W14C3 |
| $PM_{2.5}$ | 28.2 | 21.5 | 27.0 | 26.0 | 22.7 | 30.2 | 31.7 | 31.3 | 33.4 | 37.6 | 38.6 | 43.5 | 38.3 |
| $nss\text{-}K^+$ | 1.4 | 0.4 | 0.6 | 0.5 | 1.0 | 0.4 | 0.5 | 0.2 | 0.4 | 0.9 | 0.4 | 0.6 | 0.5 |
| $nss\text{-}Mg^{2+}$ | 0.4 | 0.4 | 0.4 | 0.5 | 0.4 | 0.5 | 0.5 | 0.5 | 0.5 | 0.5 | 0.6 | 0.6 | 0.4 |
| $nss\text{-}Ca^{2+}$ | 0.2 | 0.3 | 0.4 | 0.3 | 0.5 | 0.3 | 0.4 | 0.3 | 0.5 | 0.5 | 0.4 | 0.6 | 0.5 |
| $nss\text{-}SO_4^{2-}$ | 4.5 | 3.3 | 3.7 | 3.9 | 4.0 | 4.5 | 4.0 | 4.2 | 4.7 | 6.3 | 4.7 | 5.2 | 4.8 |
| $NO_3^-$ | **3.1** | **2.5** | **2.6** | **2.9** | **2.3** | **3.0** | **2.9** | **2.9** | **3.2** | **4.0** | **3.2** | **3.4** | **3.4** |
| $ss\text{-}K^+$ | 0.1 | 0.0 | 0.1 | 0.1 | 0.0 | 0.1 | 0.1 | 0.1 | 0.1 | 0.0 | 0.1 | 0.1 | 0.1 |
| $ss\text{-}SO_4^{2-}$ | 0.3 | 0.3 | 0.3 | 0.3 | 0.3 | 0.4 | 0.3 | 0.4 | 0.4 | 0.3 | 0.5 | 0.5 | 0.5 |
| $ss\text{-}Ca^{2+}$ | 0.1 | 0.0 | 0.1 | 0.1 | 0.0 | 0.1 | 0.1 | 0.1 | 0.1 | 0.0 | 0.1 | 0.1 | 0.1 |
| $ss\text{-}Mg^{2+}$ | 0.1 | 0.1 | 0.2 | 0.2 | 0.1 | 0.2 | 0.2 | 0.2 | 0.2 | 0.1 | 0.2 | 0.2 | 0.2 |
| $meas\text{-}Cl^-$ | 1.8 | 1.7 | 1.9 | 2.0 | 1.5 | 2.2 | 2.1 | 2.1 | 2.2 | 1.4 | 2.8 | 2.8 | 2.9 |
| $meas\text{-}Na^+$ | 1.2 | 1.2 | 1.3 | 1.3 | 1.0 | 1.5 | 1.3 | 1.5 | 1.4 | 1.0 | 1.9 | 2.0 | 1.9 |

Unit: $\mu g\ m^{-3}$; nss: non-sea salt ions; ss: sea salt ions; meas: measured ions
Islands represents the Xiaomen site at the Penghu Islands

**Table 5. The percentages of metallic elements, Ni/Fe, and Ni/Al in PM$_{2.5}$ sampled in Asia.**

| Sampling Locations | Sampling Periods | Ti µg m$^{-3}$ | Fe µg m$^{-3}$ | Mn µg m$^{-3}$ | Ca µg m$^{-3}$ | Mg µg m$^{-3}$ | K µg m$^{-3}$ | Al µg m$^{-3}$ | Pb µg m$^{-3}$ | Ni µg m$^{-3}$ | Cr µg m$^{-3}$ | Zn µg m$^{-3}$ | Ni/Fe | Ni/Al | References |
|---|---|---|---|---|---|---|---|---|---|---|---|---|---|---|---|
| Penghu Islands | Dec 2013-Dec 2014 | 1.2±0.6 | 1.3±0.4 | 0.7±0.3 | 2.0±0.9 | 2.3±1.2 | 2.2±1.6 | 3.0±1.2 | 0.7±0.1 | 0.9±0.3 | 0.6±0.1 | 0.6±0.2 | 0.7±0.3 | 0.3±0.1 | This study |
| Taiwan Strait | Dec 2013-Dec 2014 | 1.1±0.3 | 0.8±0.2 | 0.8±0.2 | 2.2±1.2 | 2.4±1.3 | 2.3±0.9 | 2.0±1.0 | 0.8±0.1 | 0.7±0.1 | 0.5±0.1 | 0.4±0.1 | 0.9±0.4 | 0.4±0.1 | This study |
| West-side of Taiwan Strait | June 2013-April-2015 | 1.1±0.4 | 0.9±0.3 | 0.6±0.1 | 2.0±1.0 | 1.9±0.9 | 2.5±1.3 | 1.2±0.5 | 0.6±0.3 | 0.6±0.2 | 0.4±0.2 | 0.7±0.4 | 0.7±0.2 | 0.5±0.2 | Li et al., 2016 |
| East-side of Taiwan Strait | June 2013-April-2015 | 1.0±0.3 | 1.0±0.5 | 0.7±0.2 | 2.2±1.3 | 2.4±1.3 | 3.1±1.1 | 1.1±0.6 | 0.7±0.3 | 0.7±0.2 | 0.6±0.1 | 0.7±0.3 | 0.7±0.3 | 0.6±0.2 | Li et al., 2016 |
| Over Sea of Taiwan Strait | June 2013-April-2015 | 1.2±0.2 | 1.0±0.5 | 0.8±0.2 | 2.8±1.7 | 3.1±1.8 | 4.5±2.0 | 1.2±0.3 | 1.0±0.3 | 0.8±0.2 | 0.9±0.3 | 0.8±0.4 | 0.8±0.3 | 0.7±0.2 | Li et al., 2016 |
| East China Sea | Spring 2011 | 0.8 | 5.3 | 0.0 | 6.5 | 2.9 | - | 12.4 | 0.0 | 0.0 | - | 0.2 | - | - | Zhao et al. 2015 |
| Taipei | Spring 2000 | 0.2 | 0.0 | 0.1 | 2.0 | 0.6 | - | 3.0 | 0.1 | - | - | 0.3 | - | - | Hsu et al., 2004 |
| Bohai Sea | 2006-2007 | 0.0 | 0.1 | 0.0 | 4.7 | 3.1 | - | 1.1 | 0.0 | - | - | 0.0 | - | - | Ji et al., 2011 |
| East China Sea | Spring 2008 | 0.1 | 0.8 | 0.0 | 1.4 | 0.8 | - | 0.8 | 0.0 | - | - | 0.9 | - | - | Wu et al., 2010 |

**Table 6. Comparison of OC and EC around the Taiwan Strait and East China Sea.**

| City | Location | Sampling Periods | $PM_{2.5}$ µg m$^{-3}$ | TC µg m$^{-3}$ | OC µg m$^{-3}$ | EC µg m$^{-3}$ | OC/EC | References |
|------|----------|------------------|------------|------|------|------|-------|-----------|
| Xiamen | | 2011-2013 | 51.5±20.4 | 10.0 | 8.7±4.0 | 1.3±0.2 | 6.6 | Wu et al., 2015 |
| Quanzhou | | 2011-2013 | 53.9±14.8 | 10.5 | 10.1±3.0 | 1.5±0.3 | 6.8 | Wu et al., 2015 |
| Putian | Southeast coastline | 2011-2013 | 50.3±14.4 | 9.3 | 8.1±2.0 | 1.2±0.2 | 7.0 | Wu et al., 2015 |
| Fuzhou | of China | 2011-2013 | 58.7±14.8 | 13.8 | 12.0±2.9 | 1.8±0.4 | 4.2 | Wu et al., 2015 |
| Xiamen | | 2009-2010 | 72.1±34.2 | 22.6 | 19.3±9.5 | 3.3±1.3 | 5.8 | Zhang et al., 2011 |
| Fuzhou | | 2007-2008 | 44.3±16.3 | 10.7 | 8.50 | 2.2 | 3.9 | Xu et al., 2012 |
| Xiamen | | 2013-2015 | 60.8±23.7 | 10.5 | 7.9 | 2.6 | 3.0 | Li et al., 2016a |
| Fuzhou | | 2013-2015 | 56.0±16.3 | 10.1 | 8.0 | 2.1 | 3.5 | Li et al., 2016a |
| Kaohsiung-Inland | Southwestern coastline of Taiwan Strait | 2006-2009 | 56.5 | 7.3 | 5.4 | 1.9 | 2.9 | Tsai et al., 2010 |
| Kaohsiung-offshore | | 2006-2009 | 46.1 | 5.2 | 3.4 | 1.8 | 1.9 | Tsai et al., 2010 |
| Kaohsiung | Southwestern coastline of Taiwan Strait | 2013-2015 | 43.8±19.6 | 7.6 | 5.2 | 2.4 | 2.2 | Li et al., 2016a |
| Taichung | Central coastline of Taiwan Strait | 2013-2015 | 37.5±11.3 | 6.7 | 4.5 | 2.2 | 2.1 | Li et al., 2016a |
| | | 2003-2007 | 47.9±2.0 | 13.2 | 9.5±0.4 | 3.7±0.1 | 2.6 | Chou et al., 2010 |
| New Taipei City | North coastline of Taiwan Strait | 2013-2015 | 26.1±6.7 | 5.0 | 3.6 | 1.4 | 2.6 | Li et al., 2016a |
| Cap Fuguei | | 2003-2007 | 28.0±0.9 | 5.1 | 3.8±0.2 | 1.3±0.1 | 2.9 | Chou et al., 2010 |
| Penghu islands | Islands on Taiwan Strait | 2003-2007 | 23.3±0.7 | 3.0 | 2.2±0.2 | 0.8±0.0 | 2.8 | Chou et al., 2010 |
| | | 2013-2015 | 26.3±6.2 | 4.4 | 3.2 | 1.2 | 2.6 | Li et al., 2016a |
| | | 2013-2015 | 29.5±6.2 | 5.6±1.3 | 4.0±0.9 | 1.6±0.3 | 2.6 | In this study |
| Over sea | Taiwan Strait | 2013-2015 | 32.2±6.3 | 4.8±1.2 | 3.4±0.9 | 1.4±0.3 | 2.4 | In this study |

**Table 7. Comparison of chemical composition ratios for PM$_{2.5}$ around the Taiwan Strait and East China Sea.**

| Sampling Sites | Location | EC/TC | K$^+$/TC | NO$_3^-$/nss-SO$_4^{2-}$ | TC/SO$_4^{2-}$ | Reference |
|---|---|---|---|---|---|---|
| Xiamen | Southeast coastline of China | 0.25±0.01 | 0.16±0.08 | 0.57±0.11 | 0.96±0.15 | Li et al., 2016a |
| Fuzhou | Southeast coastline of China | 0.20±0.02 | 0.14±0.09 | 0.68±0.03 | 1.25±0.27 | Li et al., 2016a |
| Penghu | Islands on Taiwan Strait | 0.26±0.04 | 0.22±0.05 | 0.59±0.06 | 1.07±0.12 | Li et al., 2016a |
| Kaohsiung | Southwestern coastline of Taiwan Strait | 0.31±0.02 | 0.11±0.07 | 0.52±0.13 | 0.98±0.13 | Li et al., 2016a |
| Taichung | Central coastline of Taiwan Strait | 0.32±0.02 | 0.12±0.07 | 0.64±0.06 | 1.05±0.09 | Li et al., 2016a |
| Taipei | North coastline of Taiwan Strait | 0.27±0.04 | 0.13±0.10 | 0.63±0.09 | 1.06±0.16 | Li et al., 2016a |
| Fuzhou | Southeast coastline of China | 0.2 | 0.02 | 0.41 | 0.99 | Xu et al., 2012 |
| Xiamen | Southeast coastline of China | 0.17 | 0.05 | 0.47 | 1.17 | Zhang et al., 2012 |
| Xiamen | Southeast coastline of China | 0.18 | - | 0.45 | 1.19 | Zhao et al., 2015 |
| Qingdao | Southeast coastline of China | 0.19 | 0.09 | 0.95 | 1.54 | Cao et al., 2012 |
| Hong Kong | Southeast coastline of China | 0.34 | 0.04 | 1.02 | 0.94 | Cao et al., 2012 |
| Shanghai | Central coastline of China | 0.24 | 0.04 | 1.03 | 1.51 | Cao et al., 2012 |
| Seoul, Korea | Korea | 0.25 | 0.03 | 0.87 | 1.72 | Heo et al., 2008 |
| Yokohama | Southeast coastline of Japan | 0.34 | 0.02 | 1.07 | 1.5 | Khan et al., 2010 |
| W13 | Taiwan Strait | 0.30±0.02 | 0.17±0.05 | 0.71±0.03 | 0.99±0.12 | In this study |
| S14 | Taiwan Strait | 0.29±0.01 | 0.13±0.07 | 0.66±0.05 | 0.94±0.05 | In this study |
| W14 | Taiwan Strait | 0.28±0.01 | 0.10±0.02 | 0.67±0.02 | 1.18±0.08 | In this study |

**Table 8. The distribution of major chemical components of PM$_{2.5}$ at the coastal sites around the Taiwan Strait and East China Sea.**

| Sampling Sites | Location | Average Conc. | SO$_4^{2-}$ | NO$_3^-$ | NH$_4^+$ | Organic Materials | Crustal Materials | Elemental Carbon | Sea Salt | References |
|---|---|---|---|---|---|---|---|---|---|---|
| Xiamen | Southeast coastline of China | 59.2 | 11.1 | 6.0 | 4.8 | 12.7 | 12.6 | 2.5 | 4.5 | Li et al., 2016a |
| Fuzhou | Southeast coastline of China | 53.6 | 8.0 | 5.3 | 4.2 | 12.8 | 14.2 | 2.0 | 4.9 | Li et al., 2016a |
| Penghu | Islands on Taiwan Strait | 23.6 | 3.9 | 2.1 | 0.8 | 4.6 | 7.8 | 1.1 | 2.1 | Li et al., 2016a |
| Kaohsiung | Southwestern coastline of Taiwan Strait | 41.8 | 7.6 | 3.7 | 1.9 | 8.2 | 10.4 | 2.3 | 3.7 | Li et al., 2016a |
| Taichung | Central coastline of Taiwan Strait | 35.6 | 5.9 | 3.5 | 1.5 | 6.9 | 8.7 | 2.0 | 2.6 | Li et al., 2016a |
| Taipei | North coastlineof Taiwan Strait | 24.9 | 4.4 | 2.6 | 1.0 | 5.5 | 7.0 | 1.3 | 2.2 | Li et al., 2016a |
| Xiamen | Southeast coastline of China | 74.2 | 12.6 | 7.4 | 6.7 | 25.2 | 13.4 | 5.2 | - | Cao et al., 2012 |
| Shanghai | Central coastline of China | 139.4 | 20.9 | 16.7 | 13.9 | 44.6 | 29.3 | 8.4 | - | Cao et al., 2012 |
| Qingdao | Southeast coastline of China | 134.8 | 21.6 | 18.9 | 14.8 | 41.8 | 24.3 | 5.4 | - | Cao et al., 2012 |
| Hong Kong | Southeast coastline of China | 88.4 | 21.2 | 9.7 | 7.1 | 22.1 | 15.9 | 6.2 | - | Cao et al., 2012 |
| Seoul | Korea | 37.6 | 5.8 | 5.2 | 3.7 | 11.4 | 2.8 | 2.9 | - | Heo et al., 2008 |
| Yokohama | Southeast coastline of Japan | 37.6 | 6.9 | 1.8 | 4.1 | 8.2 | - | 3.5 | - | Khan et al., 2010 |
| W13 | Taiwan Strait | 25.7 | 4.2 | 2.8 | 1.1 | 4.7[†] | 5.4[*] | 1.2 | 3.7[※] | In this study |
| S14 | Taiwan Strait | 29.8 | 4.6 | 2.9 | 1.8 | 5.0[†] | 7.2[*] | 1.3 | 4.0[※] | In this study |
| W14 | Taiwan Strait | 39.5 | 5.7 | 3.5 | 2.4 | 7.6[†] | 8.7[*] | 1.9 | 4.9[※] | In this study |

5   Unit: μg m$^{-3}$; OM[†]=1.6*OC (in this study); CM[*]=12*[Al] (in this study); Sea Salt[※]=[Cl$^-$]+1.47*[Na$^+$] (in this study).

**Table 9. Comparison of chloride deficit measured in marine aerosols at the coastal regions and in the remote oceans.**

| Regions/Seasons | Sample Size Range | Cl- Deficit Percentages | References |
|---|---|---|---|
| South China Sea/winter | Coarse and fine aerosols | 86% and 98% for fine-mode; 29% and 30% for coarse-mode | Hsu, et al., 2007 |
| East China Sea/(Korea) spring during Asian dust periods (ADS) and non-Asia dust periods (NADS) | Coarse and fine aerosols | 40% for fine-mode and 13% for coarse-mode during ADS periods, while during NADS periods 55% for fine-mode and 16% for coarse-mode | Park et al., 2004 |
| East coast of US/spring | Size-segregated aerosols | 14% | Keene and Savole, 1998 |
| Tropical northern Atlantic ocean/ spring | Coarse and fine aerosols | 29.7±9.9% for fine-mode and 11.9±13.3% for coarse-mode. | Johansen et al., 2000 |
| Northern Indian Ocean/ late spring and summer | SW monsoon and inter-monsoon period | 3.5±6.3% in SW monsoon and 15.0±9.0 in the inter-monsoon periods | Johansen et al., 1999 |
| Tropical Arabian Sea/spring | Coarse and fine aerosols | 89±9% for fine-mode and 25.6±21.3% for coarse-mode. | Johansen and Hoffmann, 2004 |
| NW Mediterranean Sea | Size-segregated aerosols | 18.5±14.5% | Sellegri et al., 2001 |
| Southwestern coastal area of Taiwan Strait/ inland and offshore areas | Coarse and fine aerosols | 33.8±9.7% on inland and 22.5±6.9% on offshore for fine-mode; 33.8±9.1% on inland and 15.4±3.1% on offshore for coarse-mode. | Tsai et al., 2010 |
| West-side, east-side, offshore areas and sampling boat on sea of Taiwan Strait (2013-2015) | Fine aerosols | 40.1% on west-side, 41.2% on east-side, 27.8% on offshore area, and 23.5% on sea | This study |
| Offshore and sea samples at the Taiwan Strait/ winter and spring | Fine aerosols | 16.20% to 19.19% at offshore site, 7.56% to 20.41% over sea | This study |
| Coastal Antarctica | Size-segregated aerosols | 10–20% | Jourdain and Legrand, 2002; Rankin and Wolff, 2003 |

[Figure]

**Figure 1. The navigation routes and courses during three cruise sampling campaigns in the Taiwan Strait.**

[Figure]

**Figure 2. Comparison of the the chemical species of PM$_{2.5}$ analyzed with IC and ICP-AES.**

[Figure]

**Figure 3. The concentrations of sea salt- and non-sea salt-water-soluble ions and their contribution to PM$_{2.5}$ sampled during the cruise sampling campaigns.**

[Figure]

20 **Figure 4. The percentages of non-sea salt water-soluble ionic species in the measured water-soluble ionic species of PM$_{2.5}$ sampled over sea and at the offshore islands during the cruise sampling campaigns.**

[Figure]

**Figure 5. The variation of estimated concentrations of OC, EC and OC/EC ratios in PM$_{2.5}$ over sea and at the offshore islands during the cruise sampling campaigns.**

[Figure]

**Figure 6. The enrichment factors of metallic elements in PM$_{2.5}$ over Sea and at the Offshore Islands.**

[Figure]

Figure 7. The chloride deficit and the molar ratio of Cl⁻/Na⁺ in $PM_{2.5}$ over sea and at the offshore islands during the cruise sampling campaigns.